



# Assimilation of probabilistic flood maps from SAR data into a hydrologic-hydraulic forecasting model: a proof of concept.

Concetta Di Mauro[1,2], Renaud Hostache[1], Patrick Matgen[1], Ramona Pelich[1], Marco Chini[1], Peter Jan van Leeuwen[2,4], Nancy Nichols[2], and Günter Blöschl[3]

[1]Luxembourg Insitute of Science and technology
[2] University of Reading,UK
[3]Vienna University of Technology
[4]Department of Atmospheric Science, Colorado State University, USA

**Correspondence:** Concetta Di Mauro (concetta.dimauro@list.lu)

**Abstract.** Coupled hydrologic and hydraulic models represent powerful tools for simulating streamflow and water levels along the riverbed and in the floodplain. However, input data, model parameters, initial conditions and model structure represent sources of uncertainty that affect the reliability and accuracy of flood forecasts. Assimilation of satellite-based Synthetic Aperture Radar observations into a flood forecasting model are generally used to reduce such uncertainties. In this context, we
evaluate how sequential assimilation of flood extent derived from synthetic aperture radar data can help in improving flood forecasts. In particular, we carried out twin experiments based on a synthetically generated data-set with controlled uncertainty. To this end, two assimilation methods are explored and compared: the Sequential Importance Sampling (standard method) and its enhanced method where a tempering coefficient is used to inflate the posterior probability (adapted method) and to reduce degeneracy. The experimental results show that the assimilation of SAR probabilistic flood maps significantly improves the
predictions of streamflow and water elevation, thereby confirming the effectiveness of the data assimilation framework. In addition, the assimilation method significantly reduces the spatially averaged root mean square error of water levels with respect to the case without assimilation. The critical success index of predicted flood extent maps is significantly increased by the assimilation. While the standard method proves to be more accurate in estimating the water levels and streamflow at the assimilation time step, the adapted method enables a more persistent improvement of the forecasts. However, although the use
of a tempering coefficient reduces the degeneracy problem, the accuracy of model simulation is lower at the assimilation time step.

# 1   Introduction

Floods represent one of the major natural disasters with a global annual average loss of US $104 billion (UNISDR, 2015).
Extent of flood damages have risen during the last years due to climate-driven changes and an increase in the asset values





of floodplains (Blöschl et al., 2019). This emphasizes the need for reliable and cost-effective flood forecasting models to predict flood inundations in near real-time. Hydrologic and hydraulic models represent useful tools for simulating flood extent, discharge and water levels in the river bed and on the floodplain. However, both the models and the observations used as inputs for running, calibrating and evaluating the models are affected by uncertainty.

Data assimilation (DA) aims at improving model predictions by updating model states and/or parameters based on observations (Moradkhani et al., 2005). It optimally combines observations with the system state derived from a numerical model accounting for both model and observation errors. Ideally, *in situ* data are systematically assimilated into flood forecasting models, but these observations are not always available (e.g. in *ungauged* catchments) and only provide space-limited information (Grimaldi et al., 2016). Therefore, satellite Earth Observation (EO) data, and in particular Synthetic Aperture Radar (SAR)

images, represent a valuable complementary dataset to *in situ* observations due to their capacities to provide frequent updates of flooded areas at a large scale. In addition, as the corresponding EO data archives are growing fast, historical observational data spanning an extended period of time can be assimilated into large scale hydrodynamic models. SAR sensors are able to acquire images of flooded areas and permanent water bodies during day and night almost regardless of weather conditions. The backscattered signal depends on the dielectrical properties of the imaged objects. Smooth surfaces, such as open water bodies,

interact with the transmitted pulse so that a very limited part of the signal is backscattered to the satellite resulting in dark areas in the acquired image. Different information about water extent can be extracted from a SAR image and used to improve the forecasts using DA techniques. For instance, several studies have used EO-derived water levels to improve flood forecasts [e.g. Andreadis et al. (2007), García-Pintado et al. (2015), Matgen et al. (2010), Revilla-Romero et al. (2016), Giustarini et al. (2011), Hostache et al. (2010)]. The water levels are estimated by merging pre-selected flood extent limits extracted from the

SAR satellite imagery with a digital elevation model (DEM). This step requires precise flood contour maps and high resolution DEMs which are not always available (Hostache et al., 2018).

    So far, in the existing literature, only a few studies have used used Kalman Filter (KF), Four-Dimensional Variational (4DVar) and Particle Filter (PF) techniques for assimilating flood maps into flood forecasting models [Lai et al. (2014), Revilla-Romero et al. (2016), Cooper et al. (2018), Cooper et al. (2019), Hostache et al. (2018)]. Lai et al. (2014) have assimilated the surface

water extent extracted from 250 m resolution MODIS data via the 4DVar. Revilla-Romero et al. (2016) have used the ensemble Kalman filter (EnKF), which is a variant of the KF where an ensemble of state vectors are drawn from the distribution of the state to repres, to Revilla-Romero et al. (2016) assimilate the information of surface water extent from the GFDS (Global Flood Detection System) data with a resolution of 10 km. The main difficulty of assimilating flood extent is due to the fact that the latter is not a state variable of the model since it only represents the set of water pixels of the satellite image. Therefore, in

these studies the backscatter information needs to be transformed into state variables (either discharge or water levels) in order to be assimilated.

    Cooper et al. (2019) have used an EnKF to update a 2D hydrodynamic model. In this case, the backscatter values are not transformed into state variables of the system. Instead, the simulated dry and wet pixels are converted into equivalent SAR backscatter values corresponding to the spatial mean of the wet and dry pixels of the SAR observation, at a given time, as

defined by the method proposed by Giustarini et al. (2016) . Cooper et al. (2019) compared the SAR backscatter-based assimi-




lation method with the flood edge assimilation method and showed that the new observation operator performs well compared to the assimilation of flood-edge water elevation observations.

Even though 4DVar and KFs may account for the non-linearity of the system evolution, the probability density funtions (PDFs) of the observations and of the model errors are still considered Gaussian and the analysis step is built-up on iterative lineariza-

tions of model equations and observation operators (van Leeuwen, 2010). PFs have the advantage of relaxing the assumption that PDFs of both the observation and model errors are Gaussian (Moradkhani et al., 2005). Hostache et al. (2018) used a variant of the PF with Sequential Importance Sampling (SIS), to assimilate probabilistic flood maps (PFMs) derived from SAR data into a coupled hydrologic-hydraulic model with the assumption that the rainfall is the main source of uncertainty. Their study showed that the assimilation of PFMs is beneficial: the number of correctly predicted flooded pixels increases as

compared to the case without any assimilation, hereafter called *Open Loop* (OL). Forecast errors are reduced by a factor of 2 at the assimilation time and improvements persist for subsequent time steps up to 2 days. However, the improvements are not systematic: for some cases the updated hydraulic output deviates from the observations.

The reason for such outliers could be the assumption that rainfall represents the dominating source of uncertainty together with satellite observation errors, excluding other possible sources of uncertainty in the model system. Even though the assump-

tion seems to be rather realistic and suitable in operational cases, given that rainfall uncertainty has been generally identified as one of the major causes of poorly performing models [Koussis et al. (2003), Pappenberger et al. (2005), Komma et al. (2008), Nester et al. (2012)], coupled models may have additional sources of uncertainty affecting the results. In this context, in this study we carry out a similar experiment but this time in a controlled environment so that rainfall is actually the only source of uncertainty. Therefore, the objective of this study is to further assess and validate the proposed framework when the underlying

assumptions are respected.

Moreover, Hostache et al. (2018) highlighted that degeneracy may be a major issue of PFs: after the assimilation, the number of particles with high weights reduces to few or one particle so that the ensemble loses statistical significance. To overcome this issue Hostache et al. (2018) used a site-dependent tempering coefficient which inflates the posterior probability. In our study, we propose to adopt an enhanced tempering coefficient. The latter is a function of the desired effective ensemble size after the

assimilation. The adapted method is compared to the standard method where only one particle is left after the assimilation.

The detection of flooded areas in SAR images could be rather straightforward. However, for particular cases the SAR backscattering values of flooded and non-flooded areas are difficult to distinguish, leading to a wrong classification in the flood mapping results. Such errors could be due to particular atmospheric conditions (e.g. wind, snow, precipitation), to water-look like areas (e.g. asphalt, sand, shadow), to obstructing objects (e.g. dense canopy, buildings) or errors inherent to coherent imaging sys-

tems (e.g. speckle) as mentioned in (Giustarini et al., 2015). Detecting and removing these errors represent one of the main scientific challenges of using SAR data for the systematic, fully automated, and large-scale flood monitoring. In Hostache et al. (2018), speckle errors are taken into account through the Bayesian approach of Giustarini et al. (2016) but no conclusions are given on the effect of misclassified pixels in the SAR observations. Consequently, in this synthetic experiment, bias is added within the SAR image with the goal of understanding how robust the proposed method is with respect to the proportion of

misclassified SAR pixels.





The objective of the present study is to assess the main strengths and limitations of a previously proposed DA framework in a fully controlled environment via synthetic twin experiments in order to draw more more general and comprehensive conclusions. A sensitivity analysis of the DA framework with respect to the tempering coefficient is conducted. Results are evaluated not only locally but also over the entire flood domain and for subsequent time steps after assimilation.

## 2 Methods


The proposed methodology is based on numerical experiments conducted with synthetically generated data as illustrated in the flow chart given in Figure 1. In this framework, the following data inputs and models are employed:

1. *True* rainfall time series are used to generate the *true* hydrologic-hydraulic model simulation.

2. Synthetic SAR observations are generated from the *true* model run (i.e. from the simulated flood extent map).


3. The *true* rainfall time series are randomly perturbed to obtain an ensemble of upstream boundary discharges via the hydrologic model. The simulated discharge data are then used to realize an ensemble of hydraulic model runs.

4. The synthetic SAR observations are assimilated into the coupled hydrologic-hydraulic model via a SIS-based variant of the Particle Filter (PF).

### 2.1 Coupled hydrologic-hydraulic model: synthetic truth and ensemble


The coupled modelling system consists of a hydrologic model coupled with a hydraulic model. The hydrologic model is used to compute the run-off at the upstream boundaries of the hydraulic model. The hydrologic model used in this study is SUPER-FLEX which is a framework for conceptual hydrologic modelling introduced by Fenicia et al. (2011). The model structure is a combination of generic components: reservoirs, connection elements and lag functions. In this study, a lumped conceptual model and its structure as a combination of three reservoirs are used: an unsaturated soil reservoir with storage $S_{UR}$, a fast


reacting reservoir with storage $S_{FR}$, and a slow reacting reservoir with storage $S_{SR}$. A lag function has been added at the outlet of the slow and fast reacting reservoirs. The hydraulic model is based on LISFLOOD-FP (Bates and Roo, 2000; Neal et al., 2012) and simulates flood extent, water levels and streamflows along the river and on the floodplain. A sub-grid 1D kinematic solver is used for the channel flow. When the storage capacity of the river is exceeded, the water spills into the floodplain and a 2D diffusion wave scheme neglecting the convective acceleration is used for the floodplain flow simulation. Channel width,


channel depth, slope of terrain, friction of the flood domain and channel bathymetry are defined in each cell of the model domain as described in Wood et al. (2016). A uniform flow condition is imposed downstream. No later inflow in the hydraulic model is assumed.

The *true* meteorological data (i.e., temperature and rainfall) are used as input of the hydrologic-hydraulic model to simulate streamflow and water level time series and to provide binary maps, where each pixel is classified as flooded (with value 1) or


not flooded (with value 0), at each assimilation time step. These computational results represent the synthetic *truth* that will be



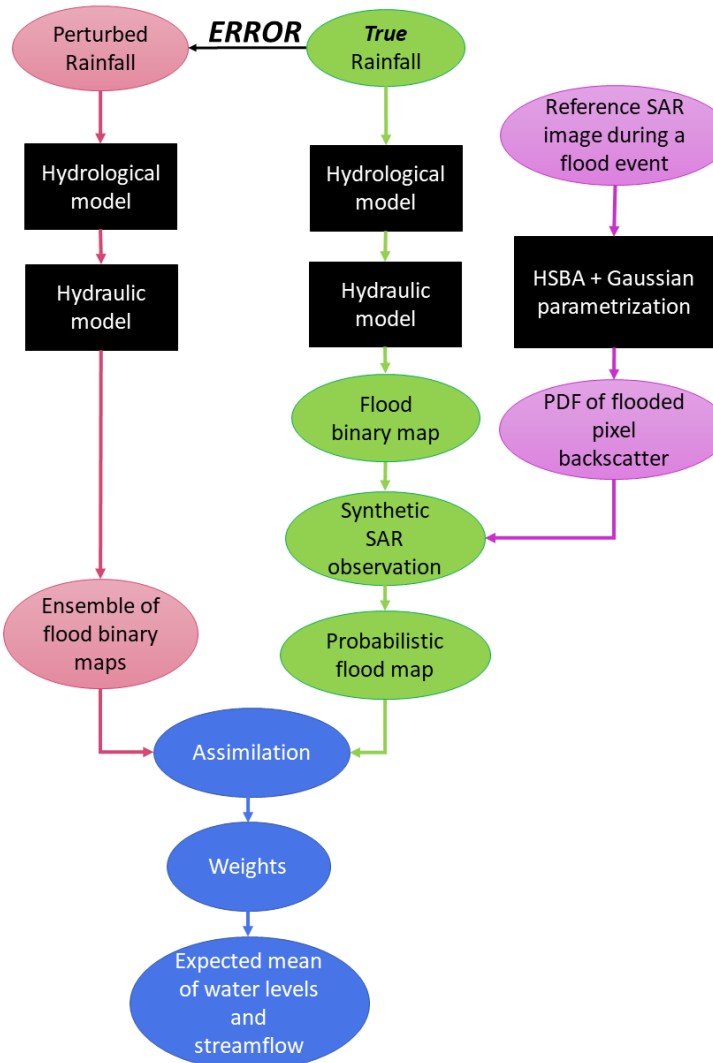

**Figure 1.** Flow chart of the synthetic experiment. The *true* rainfall is perturbed. The same flood forecasting model structure composed of a hydrological model and a hydraulic model is used to obtain the flood probabilistic map, with the use of a reference SAR image, and an ensemble of flood binary maps. The probabilistic flood map is assimilated into the ensemble of flood binary maps to obtain the weights which are then used to compute the expected mean of water levels and discharge.

used to evaluate the performance of the proposed assimilation framework. The *true* binary maps are also used to generate the synthetic SAR observations as described in the next section.



## 2.2 Synthetic observations

In the proposed synthetic experiment, we generate synthetic SAR images (as shown in the the flowchart in Figure 1) at each time step corresponding to the real acquisition plan of the Sentinel-1 satellite constellation. Similarly to the Van Wesemael (2019) study, we make use of a real SAR image, acquired during a flood event in the past, and of the LISFLOOD-FP model to generate true flood binary maps. The histogram of the SAR image backscatter values can be approximated with two Gaussian curves relative to the flooded and not flooded pixel classes. Generally, the class of flooded pixels is often represented just by a

fraction of the SAR image scenes. Therefore, to identify and characterize areas where the flooded and not flooded classes are more balanced, the hierarchical split based approach [HSBA, Chini et al. (2017)] is applied to the selected SAR image. The parameters of the Gaussian PDFs are determined by fitting the histogram values of the HSBA selected areas. Then random backscatter values, derived from the calibrated Gaussian PDFs, are associated to the pixels of the *true* binary map indicating the presence of water and no-water areas. Once the synthetic SAR images are generated, the Giustarini et al. (2016) procedure

is applied and synthetic PFMs are derived. We have adopted the latter method in order to generate synthetic observations and to determine for each pixel of a SAR image its probability to be flooded given the recorded backscatter values $p(F|\sigma^0)$. This probability is obtained via the Bayes' theorem:

$$p(F|\sigma^0) = \frac{p(\sigma^0|F)p(F)}{p(\sigma^0)} = \frac{p(\sigma^0|F)p(F)}{p(\sigma^0|F)p(F) + p(\sigma^0|\overline{F})p(\overline{F})} \tag{1}$$

In the equation 1, $p(\sigma^0|F)$ and $p(\sigma^0|\overline{F})$ represent, respectively, the probability of the backscatter values of the flooded and

non-flooded pixels, $p(F)$ is the prior probability of a pixel being flooded and $p(\overline{F})$ is the prior probability of a pixel being non-flooded. In general, the prior probabilities are unknown and Giustarini et al. (2016) proposed to use 0.5 as default value. In this synthetic experiment, the prior probability is derived from the *true* binary map as the ratio between the number of flooded pixels and the total number of pixels at each time step. The conditional probabilities are derived from the histogram of the spatial distribution of backscatter values corresponding to the synthetically generated SAR image.

SAR observations are considered unbiased in the first part of the study. The method used in this study and proposed by Giustarini et al. (2016) aims at characterizing the speckle-induced uncertainty. However, it does not consider any other particular SAR-error causes, e.g generated by atmospheric conditions or specific ground features. Therefore, areas where such errors could occur should be masked out, otherwise the estimate of SAR-based flood extent could be compromised. In the second part of the study, these kinds of errors are integrated in the synthetic SAR observations to evaluate their effect on the

DA. Specifically, the pixels along the flood edge of each particle are selected. From this set, a given number of those pixels effectively flooded in the *true* binary map are artificially corrupted so that they belong to dry pixels according to the magnitude of error introduced in the SAR observations.





## 2.3 Ensemble generation

In a PF the prior and posterior PDFs are approximated by a set of particles. Here, we hypothesize that rainfall is the only source
of uncertainties affecting the model-based flood extent simulations. Due to this reason, different rainfall time series are used
as inputs of the coupled hydrologic-hydraulic model. Each rainfall time series is obtained by perturbing the *true* rainfall time
series following the approach proposed in Hostache et al. (2018). The rainfall is perturbed with a multiplicative random noise
from a log-normal error distribution. 128 rainfall time series are obtained and forwarded in time via the hydrologic model. It is
important to note that the same hydrologic-hydraulic model in terms of structure, initial conditions and parameters is used for
all model runs.

The reliability of the rainfall ensemble is verified with the statistical metrics proposed by De Lannoy et al. (2006). According
to the verification measurement $VM_1$ in equation 4, the ensemble spread (equation 2) has to be close to the ensemble skill
(equation 3), which is the difference between the mean $\overline{x}_k$ over the N particles of the ensemble and the observation $y_k$ at time
$k$. $VM_2$ (equation 5) verifies that the *truth* is statistically indistinguishable from the random samples of the ensemble. $VM_1$
and $VM_2$ are used to assess the *quality* of the discharge ensemble at the output of the hydrologic model. $x_{k,n}$ in the equation 2
represents the value of the variable $x$ at time $k$ for each pixel $n$.

$$ensp_k = \frac{1}{N} \sum_{n=1}^{N} (x_{k,n} - \overline{x}_k)^2 \tag{2}$$

$$ensk_k = (\overline{x}_k - y_k)^2 \tag{3}$$

$$VM_1 = \frac{<ensk>}{<ensp>} \approx 1 \tag{4}$$

$$VM_2 = \frac{<ensk>}{<mse>} \approx \frac{(N+1)}{2N} \tag{5}$$

with *mse* estimated as:

$$mse_k = \frac{1}{N} \sum_{n=1}^{N} (x_{k,n} - y_k)^2 \tag{6}$$

## 2.4 Data assimilation framework

The data assimilation framework consists of two main steps: *prediction*, i.e model simulations, and *analysis*, i.e update of
particle probabilities when an observation is available. The prior probability of the model state $x$ at a given time $k$ is represented
by a set of N independent random particles $x_n$ sampled from the prior probability distribution *p(x)* as:

$$p(x) = \frac{1}{N} \sum_{n=1}^{N} \delta(x - x_n) \tag{7}$$





In this study, the prior probability distribution is supposed to be uniform. The observations of flooded/not flooded pixels are related to the true state $x^t$ according to the following equation:

$$y = H[x^t] + \epsilon \tag{8}$$

where H is the observation operator that maps the state vector into the observation space and $\epsilon$ represents the observation errors. According to Bayes' theorem, the observations are assimilated by multiplying the prior PDF and the likelihood which is the probability density of the observation given the model state, resulting in:

$$p(x \mid y) = \frac{p(y \mid x)}{p(y)} p(x) \tag{9}$$

that is the posterior probability, i.e. the probability density function of the model state given the observations. By inserting equation 7 into equation 9 we obtain the following formula:

$$p(x \mid y) \approx \sum_{n=1}^{N} W_n \delta(x - x_n) \tag{10}$$

where $W_n$ represents the relative importance in the probability density function (i.e. global weight) given by:

$$W_n = \frac{p(y \mid x_n)}{N \cdot p(y)} = \frac{p(y \mid x_n)}{N \cdot \int p(y \mid x) p(x) \delta x} \approx \frac{p(y \mid x_n)}{\sum_j p(y \mid x_j)} \tag{11}$$

In this study, the likelihood (global weight, $W_i$) is represented by the product of the pixel-based likelihood (local weight, $w_i$), assuming the L pixel observation errors to be independent from each other.

At time k of the observation, local weights $w_{i,n}$ are defined for each particle $n$ and for each pixel i according to Hostache et al. (2018):

$$w_{i,n} = p_i(F \mid \sigma_0) M_{i,n} + [1 - p_i(F \mid \sigma_0)](1 - M_{i,n}) \tag{12}$$

$w_i, n$ is equal to the probability of a pixel being flooded as derived from the synthetically generated SAR image if the model prediction $M_{i,n}$ is "1" (flooded), whereas if the model prediction $M_{i,n}$ is equal to "0" (non-flooded), $1 - p_i(F \mid \sigma_0)$ equals the probability of a pixel not being flooded according to the observations. By doing so we assign higher probabilities to those pixels where model predictions and observations agree. Next, $W_n$ is estimated for each particle. It is computed with the normalization of the weights ensuring that the sum is equal to 1 (equation 13, *standard method*).

$$W_n = \frac{\prod_{i=1}^{L} w_{i,n}}{\sum_{n=1}^{N} \prod_{i=1}^{L} w_{i,n}} \tag{13}$$

The global weights are used to compute the expectation of the streamflows (Q) and water levels (h) at time ($k$) and per pixel ($i$) ( see equations 14, 15). We convert the model-based water depth maps into binary flood extent maps by considering that a pixel is flooded if its water level is above 10 cm.

$$\overline{h}_i = \sum_{n=1}^{N} W_n \cdot h_{i,n} \tag{14}$$





$$\overline{Q}_i = \sum_{n=1}^{N} W_n \cdot Q_{i,n} \tag{15}$$

The expectation of OL is equivalent to the mean of the ensemble because the relative importance of each particle is the same. With the application of the SIS, particles are set to the same equal weight before a new *analysis* step is performed. Unless the number of particles increases exponentially with the dimension of the system-state, the particle-filter is likely to degenerate because high probability is assigned to a single particle while all other members will result in small weights (van Leeuwen et al., 2019). PFs are often subject to degeneracy issues when due to computational reasons the number of particles is not sufficiently high (Zhu et al., 2016). After the application of the standard PF, the variance of the weights tend to increase and only a few particles of the ensemble have a non-negligible weight. To mitigate this problem, in Hostache et al. (2018), the global weight defined in the equation 13 has been adapted using a *tempering* coefficient ($\alpha$, as described by the following equation 16).

$$W_n(\alpha) = \frac{\prod_{i=1}^{L} w_{i,n}{}^{\alpha}}{\sum_{n=1}^{N} \prod_{i=1}^{L} w_{i,n}{}^{\alpha}} \tag{16}$$

Since $\alpha$ and weights have values are lower than one, adding the power of $\alpha$ in the weights formula allows for shifting all weight values closer to one. This therefore decreases the variance of the weights and inflates the posterior probability. After the assimilation, the number of particles with significant weight depends on the $\alpha$ value. The smaller $\alpha$, the higher the variance of the posterior PDF. Consequently, as argued in Hostache et al. (2018), when the $\alpha$ coefficient is small enough, this adaptation of the PF helps in reducing the degeneracy of the ensemble. The particles keep these weights to the next assimilation time. Using the tempering coefficient in this way leads to biased results because of the down-weighting of the observations by increasing their errors (van Leeuwen et al., 2019). While in the previous study by Hostache et al. (2018), the $\alpha$ value was defined so that the worst model solution would have had a non-zero global weight, in this study we propose to define $\alpha$ based on the desired effective ensemble size (EES). The previous definition of $\alpha$ was site-dependent as it relies on the number of flooded pixels, whereas EES is a measure of degeneracy based on the global weights (Arulampalam et al., 2002):

$$EES(\alpha) = \frac{1}{\sum_{n=1}^{N} (W_n(\alpha))^2} \cdot \frac{1}{N} \cdot 100 \tag{17}$$

EES is lower than $N$ and its value indicates the level of degeneracy. $\alpha$ is equal to one when the standard method is used. Decreasing the $\alpha$ coefficient leads to an increase of EES. In this study, we evaluate the effect on the DA due to variations of $\alpha$. In summary, different PFs are compared with the OL and the synthetic *truth*.: the SIS (with only a few particles from the ensemble potentially carrying non- negligible weight) and the adapted method with 5-10-20-50% EES (with the number of particles with non negligible weight increasing with the EES). It is realized that because the observation influence is down-weighted this methodology leads to slightly biased estimates. We discuss this further in the concluding section.

## 2.5 Performance metrics

To carry out the evaluation of the different assimilation scenarios, we propose to adopt the following performance metrics.





### 2.5.1 Reliability plots

Reliability diagrams are employed to statistically evaluate the synthetically generated PFMs. In such diagrams, the probability range [0;1] is subdivided into intervals of average probability $P_i$ and width $\Delta P_i$. We identify the pixels $\Omega_i$ having a probability value of $P_i \pm \Delta P_i$ in the PFM. The fraction of $\Omega_i$ pixels effectively flooded in the binary *truth* map are identified with $F_i$. The reliability diagram plots $P_i$ on the x-axis and $F_i$ on the y-axis. A reliability diagram indicating an alignment of data points close to the 1:1 line means that the PFM is statistically reliable.

### 2.5.2 Contingency maps and confusion matrix

First, we use contingency maps to graphically compare the expected flood map with the synthetic *truth* map at each assimilation time step. Pixel classification errors can be of two types: overprediction (type error I) when the pixels in the *truth* map are not flooded but are predicted as flooded, and underprediction (type error II) in the opposite case. Then, the confusion matrix numerically summarizes the results of the contingency map. It is a 2 rows by 2 columns matrix that reports the number of false positives (type I error), false negatives (type II error), true positives and true negatives.

### 2.5.3 CSI

The Critical Success Index (CSI) evaluates the goodness of fit between the *truth* map and the predicted flood extent map (Bates and Roo, 2000):

$$CSI = \frac{A}{A+B+C} \tag{18}$$

It represents the ratio between the number of pixels correctly predicted as flooded (A) over the sum of all the flooded pixels including the false positives (B, overdetection) and false negatives (C, underdetection). CSI ranges between 0 and 1 (best score).

### 2.5.4 RMSE

The root mean square error (RMSE) measures the square root of the average squared error of the predicted water levels $(h_k^p)$ against the truth $(h_k^o)$ per pixel $k$ over the total number of pixels $L$ of the flood domain.

$$RMSE = \sqrt{\frac{\sum_{i=1}^{L}\left(h_i^p - h_i^o\right)^2}{L}} \tag{19}$$

## 3 Study area and Data

Our synthetic experiment is grounded on a real test site and an actual storm event: the river Severn in the middle-west of UK (figure 2) and the July 2007 flood event, rispectively. This area has experienced several floods along the river valleys (Environment Agency, 2009) generally due to intense precipitation. While seven upstream catchments contribute to the flow along the





river Severn, in our study only one upstream catchment is considered: the Severn at the Bewdley gauging station. Our first objective is to evaluate whether the model correctly predicts the output in the simplest case, i.e. when a unique run-off input to the hydraulic model determines the flood extent and no additional contributing tributaries interfere. The ERA5 dataset (Hersbach
et al., 2019) referring to the period of July 2007 has been used in this experiment. ERA5 is a global atmospheric re-analysis dataset provided by the European Centre for Medium-Range Weather Forecasts (ECMWF). Rainfall and 2 m air temperature at a spatial resolution of approximately 25 km and a temporal resolution of 1 hour are used as input to the hydrologic model. The *true* rainfall time series is used to generate the *true* run-off before being perturbed in order to obtain 128 different particles as inputs to the hydrologic model. The boundary condition of the hydraulic model is imposed in correspondence to the red dot in Figure 2. Finally, at each time step a stack of 128 wet/dry maps is obtained. Discharges and the water levels recorded at differ-

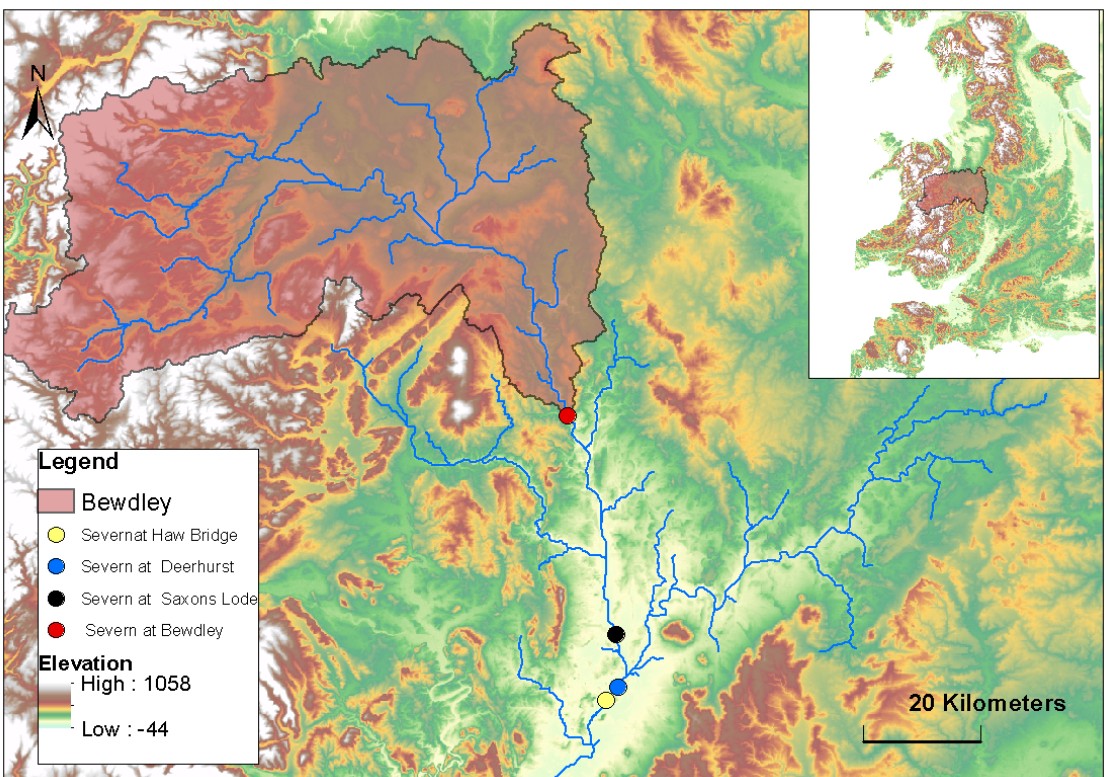

**Figure 2.** Study area: River Severn (UK). Only the boundary condition in Bewdley is taken into account. Within the sub-catchment upstream of Bewdley a lumped hydrological model is used to determine the input of the hydraulic model along the river Severn downstream. The dots represent the existing gauging stations where the performances of the DA framework are evaluated.


ent gauging stations (corresponding to the existing ones, dots in figure 2) along the river are used to evaluate the performance of the DA.




## 4    Results

The virtual satellite acquisition dates are aligned with the actual Sentinel-1 acquisition frequency. The revisit time over Europe
is around 3-4 days meaning that on average 2 satellite images are available per week. Therefore we chose to assimilate four
synthetic observations in a period of 10 days.

The prior is the probability to be flooded or not before any backscatter information is taken into account and it can be estimated
from the flood extent model output or through visual interpretation of an aerial photography in real cases. However, such
information are not always available. Therefore, Giustarini et al. (2016) set the prior probability of equation 9 to 0.5 so that
both flooded and non flooded pixels are equally likely. In this study, being based on a synthetic experiment, *true* binary flood
maps are available. Therefore, the assimilation is realized using both the estimated prior probability as the ratio between the
flooded area and the total area, and the prior probability equal to 0.5. Given the similarity of the results for both cases, in
the following sections we will discuss only the case with the estimated prior probability. The synthetic SAR observations are
shown in Figure 3. The PFMs are shown at the top and the corresponding reliability plots are provided at the bottom of Figure
5. In the reliability plots, the points align along the 1:1 line meaning that the PFMs are statistically reliable.

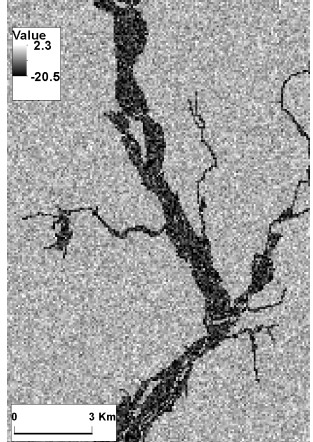
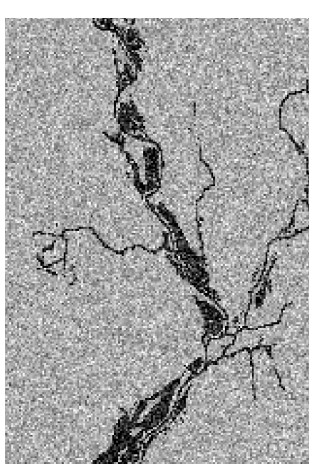
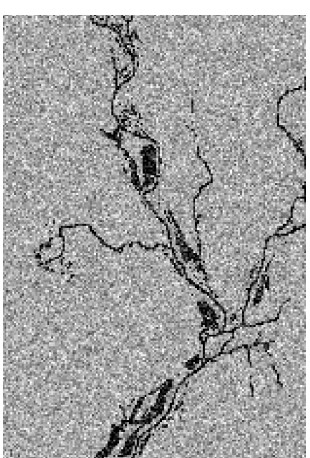
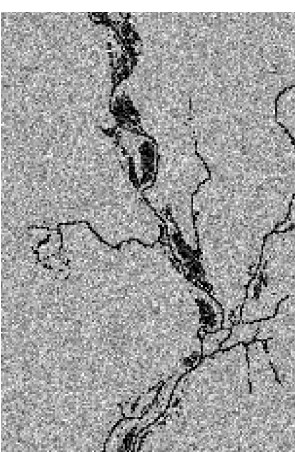

**Figure 3.** A detail of the synthetic SAR images corresponding to the 4 assimilation time steps. Darker pixels represent to lower backscatter.

The verification measurements VM$_1$ and VM$_2$ (equations 4 and 5) of the ensemble discharge in Bewdley (figure 6) are equal
to 1.047 and 0.527, respectively. These values are close to the ideal ones of 1 and 0.5.

### 4.1    CSI & RMSE at the assimilation time

At first CSI and RMSE are computed over the entire hydraulic model domain at each assimilation time step.
The general trend of the assimilation effect is positive, as errors tend to decrease at all the assimilation steps with different
assimilation methods. Even though the CSI is already high with the OL, the assimilation further improves the results and this
becomes particularly clear at the last assimilation time step. From Table 1 it can be noticed that the CSI, approximately equal to



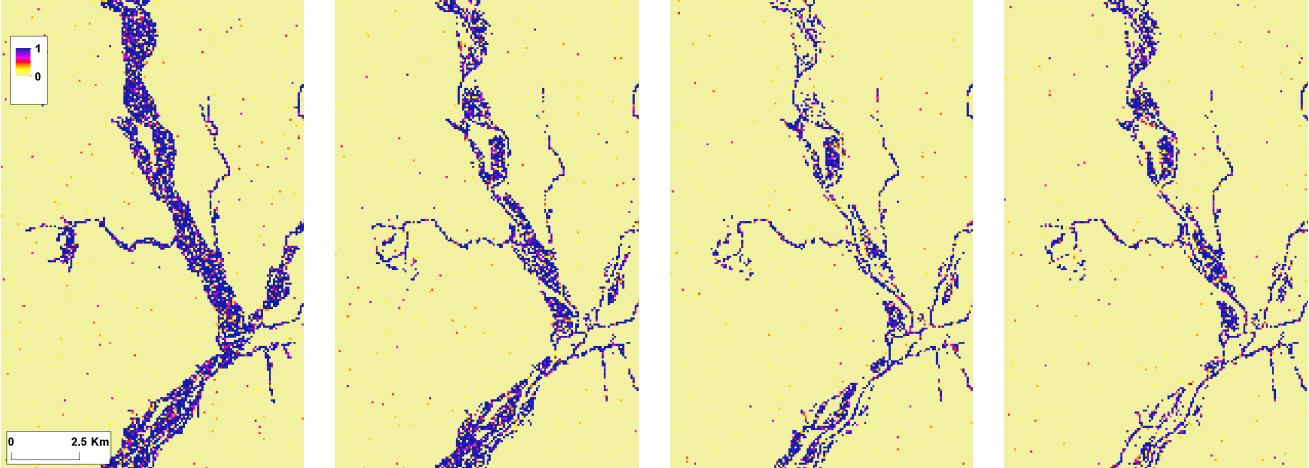

**Figure 4.** A detail of the synthetic probabilistic flood maps derived from synthetic SAR images (top). Probabilities to be flooded knowing the backscatter go from low value (yellow) to high values (blue).

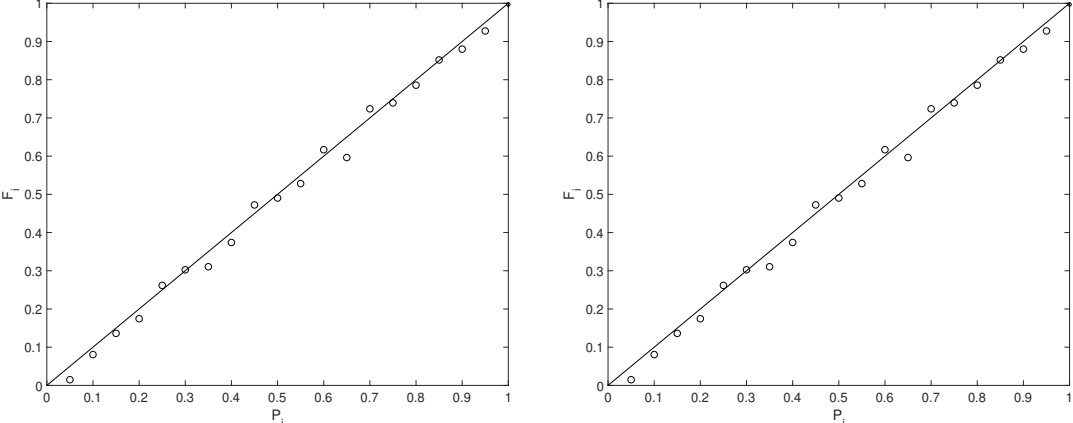

**Figure 5.** Example of the reliability plots for the verification of the synthetic probabilistic flood maps of the first two synthetic SAR images. On the x-axis probability range ($P_i$), on the y-axis the fraction of pixels within the probability range of the probabilistic flood map observed as flooded in the *true* flood binary map ($F_i$). The probabilistic flood maps are statistically reliable because the points align along the 1:1 line.

0.80 with the OL in the worst case (assimilation of the IV image), exceeds 0.96 for the different assimilation types and reaches the maximum value of 0.99 with the standard method. These results are confirmed by the RMSE reported in Table 2. Indeed, the RMSE decrease by factors larger than 2 and 3 with the standard assimilation and the 5% EES assimilation, respectively. After the $1^{st}$ assimilation, carried out close to the flood peak in Saxons Lode, the accuracy of the water level is improved by approximately 20 cm over the entire flood domain. The assimilation of the $2^{nd}$ and $4^{th}$ images has a negative effect in case the adapted method 50% EES of the assimilation particle filter is applied: the RMSE increases compared to the OL. In Figure 7,





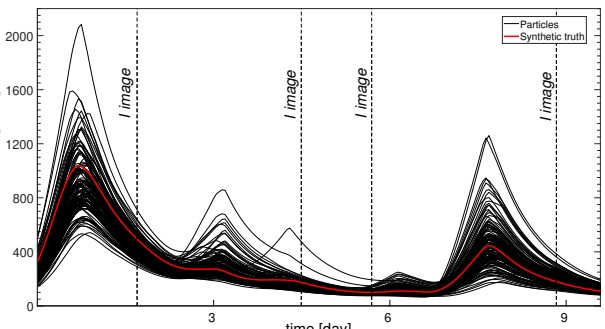 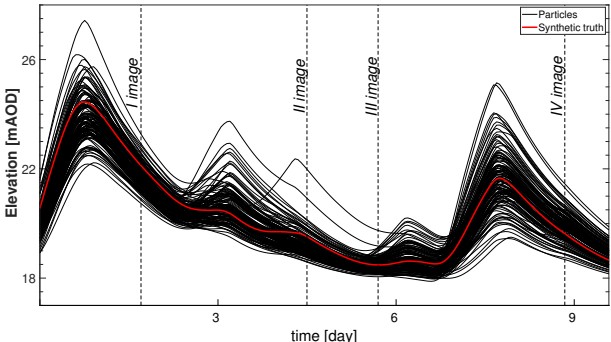

**Figure 6.** Streamflow time series (left) and water elevation time series (right) at the gauge station in Bewdley. Black lines represent the 128 particles while the red line corresponds to the synthetic truth.

**Table 1.** Critical success index values at each assimilation time step. The Open Loop where no assimilation is computed is compared with the standard method and the adapted method with an increasing effective ensemble size (EES).

| Assimilation times | Open Loop | Assimilation | | | | |
|:---:|:---:|:---:|:---:|:---:|:---:|:---:|
| | | *standard* | *5% EES* | *10% EES* | *20% EES* | *50% EES* |
| **I image** | 0.9573 | 0.9887 | 0.9914 | 0.9866 | 0.9805 | 0.9779 |
| **II image** | 0.9202 | 0.9873 | 0.9800 | 0.9758 | 0.9658 | 0.9645 |
| **III image** | 0.9437 | 0.9921 | 0.9753 | 0.9690 | 0.9622 | 0.9636 |
| **IV image** | 0.7976 | 0.9881 | 0.9754 | 0.9638 | 0.9577 | 0.9610 |

**Table 2.** Root mean square error (RMSE [m]) at each assimilation time step. The Open Loop where no assimilation is computed is compared with the standard method and the adapted method with an increasing effective ensemble size (EES).

| Assimilation times | Open Loop | Assimilation | | | | |
|:---:|:---:|:---:|:---:|:---:|:---:|:---:|
| | | *standard* | *5% EES* | *10% EES* | *20% EES* | *50% EES* |
| **I image** | 0.2608 | 0.0742 | 0.0608 | 0.0785 | 0.1501 | 0.1762 |
| **II image** | 0.1246 | 0.0526 | 0.1046 | 0.1278 | 0.1553 | 0.1704 |
| **III image** | 0.1604 | 0.0645 | 0.1103 | 0.1665 | 0.2154 | 0.2270 |
| **IV image** | 0.1702 | 0.0541 | 0.0619 | 0.1084 | 0.1899 | 0.2205 |

we provide examples of the OL and of the 5% EES approach knowing that results are similar to those of the standard method.
For each pair of images, we show on the left the results of the OL and on the right the results obtained after the assimilation.





It can be observed that in this case the OL has a tendency to over-detection; the number of red pixels is higher than the orange ones and after the assimilation the number of over-detected pixels decreases.

**I image**

Open Loop

not flooded
underdetection
overdetection
flooded

Text

Assimilation (5% EES)

**II image**

Open Loop

Text

Assimilation (5% EES)

**III image**

Open Loop

Text

Assimilation (5% EES)

**IV image**

Open Loop

Text

Assimilation (5% EES)

**Figure 7.** Contingency maps before (open loop) and after assimilation at 5 % EES at each time step. 2 types of errors can be distinguished: overdetection (red pixels) when the model predicts the pixel as flooded but the pixel is observed as not-flooded and underdetection (black pixels) when the contrary occurs. In case model and observation agree pixels are correctly classified as not-flooded (white pixels) and flooded (blue pixels).

The confusion matrix given in Table 3 provides more detail on the $4^{th}$ assimilation time step. On the one hand, the number of pixels wrongly predicted as flooded in the OL is 1196 and more than 90% of these are correctly classified as non flooded after the assimilation for both standard and 5% EES methods. On the other hand, a few pixels correctly predicted as flooded in the OL are classified not flooded after the assimilation. However, it can be argued that the number of 201 wrongly classified pixels after the assimilation is rather low compared to the 1253 pixels of the OL.





**Table 3.** Confusion matrix of the OL and of the 5% EES assimilation at the $4^{th}$ assimilation time step: OF= flooded pixels in the truth map, ON=observed non flooded in the truth map, PF= predicted flooded, PN=predicted non flooded

|  | Open Loop | | Standard | | Assimilation (5% EES) | |
|---|---|---|---|---|---|---|
|  | **PF** | **PN** | **PF** | **PN** | **PF** | **PN** |
| **OF** | 4826 | 57 | 4748 | 135 | 4815 | 68 |
| **ON** | 1196 | 264833 | 66 | 265963 | 41 | 265988 |

### 4.2 CSI & RMSE in time

The flood is simulated using an hourly time step. Consequently, it is possible to evaluate the evolution of the performance metrics CSI and RMSE in time (Figure 8). This figure shows that the OL's performance is consistently poor and the standard

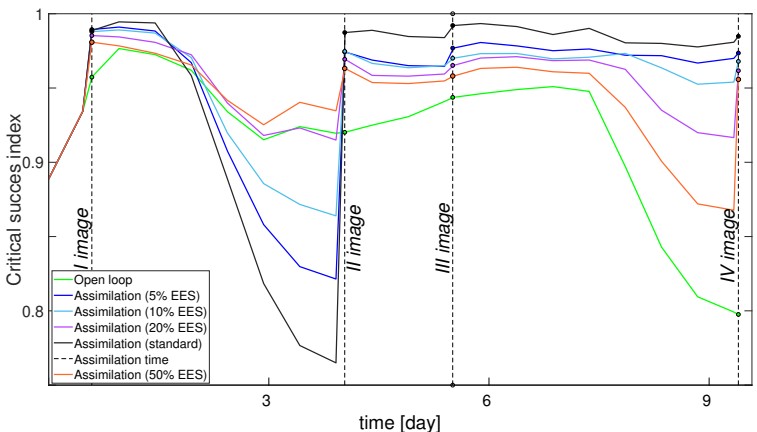

**Figure 8.** Time series of CSI of flood extent values for the different assimilation methods: open loop (green), standard assimilation (black), assimilations with 5% EES (blue), 10% EES (cyan), 20% EES (purple), 50% EES (orange).


assimilation performs best compared to the other assimilation runs at all the assimilation time steps. The assimilation runs with different EES values lie within these two extremes. It can be noted that the more particles are neglected, which is equivalent to say the lower is the EES, the higher is the performance at the assimilation time step. The reason could be linked to the fact that results are biased since not the whole tempering scheme has been followed.

Moreover, Figure 8 shows markedly different CSI time series for the different assimilation experiments. 27 hours after the first assimilation, the performances of the standard and 5% EES methods, which perform better than the other methods, start decreasing. The lowest values are reached 54 hours after the assimilation. One explanation is that the weights assigned to the particles at the $1^{st}$ assimilation time are no longer valid when hydraulic conditions change and need to be recomputed. However, things change after the $2^{nd}$ assimilation, when the performances of the standard and the 5% EES assimilation





methods remain stable until the end of the simulation time. The decrease of performances attributed to the standard and 5%
assimilation methods after the $1^{st}$ time step is due to a drastic change in the flood extent. The total number of flood pixels
reduces from 8539 to 5494 because the flood started receding. The spread of the posterior PDF with the standard and 5% EES
methods is small, meaning that only a few particles retain significant *importance* weight. Consequently, when the flood extent
changes and particles evolve in time, it may happen that the uncertainty bounds of the posterior PDF do not encompass the

*true* model state after several time steps. On the contrary, when more particles are considered (higher EES), more particles are
used to draw the posterior PDF. This gives more chances to the ensemble to encompass the synthetic *truth* and increases the
overall robustness of the method. This becomes particularly relevant when the hydraulic boundary conditions change and no
new observation is available.

As already shown in the Table 2 the standard assimilation and 5% EES predictions of water levels provide more accurate

results (figure 9). When moving away from the first assimilation, the RMSE of the best performing assimilation methods
increases. For instance, after 54 hours the RMSE of the standard method is increased by 65% compared to the RMSE of the OL.
In case different EES are considered, the RMSE values fluctuates significantly in between two assimilations and it becomes
difficult to draw any general conclusions. As the number of *important* particles increases, water levels vary significantly,
especially in the area close to the flood edge even though the flood extent does not change too much from a particle to another.

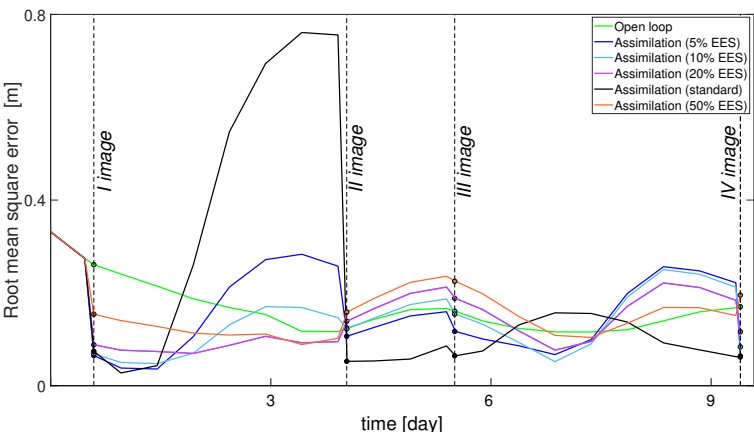

**Figure 9.** Time series of root mean square error (RMSE [m]) values for the different assimilation experiments: open loop (green), standard
assimilation (black), assimilations with 5% EES (blue), 10% EES (cyan), 20% EES (purple), 50% EES (orange).

### 4.3 Discharge and water level time series

The different assimilation runs are also compared considering the discharges and water levels at different gauge stations along
the river Severn. In the right panels of Figures 10 and 11 the different assimilation experiments are compared against the
synthetic truth (red line). In the left panels of Figures 10 and 11 the standard method and the 5% EES assimilation with the
*important* particles and the synthetic truth are shown. The plotted *important* particles represent the 5% of the ensemble with the

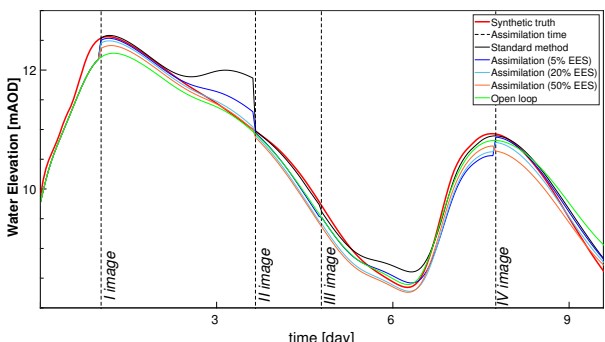
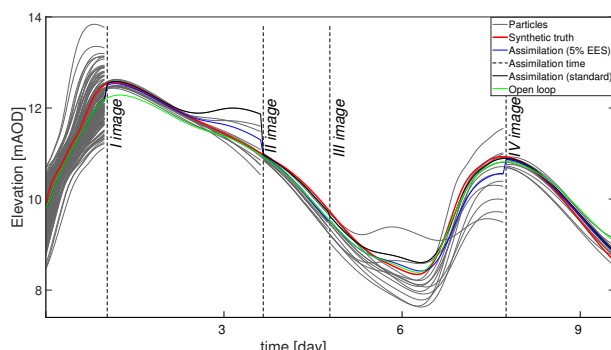

**Figure 10.** Water level time series at Saxons Lode. Left: assimilation runs with an EES of 5%(blue), 20% (cyan) and 50% (orange), OL (green), standard assimilation (black). Right: important particles after the assimilation at 5% EES (grey). Dashed lines correspond to the assimilation times.

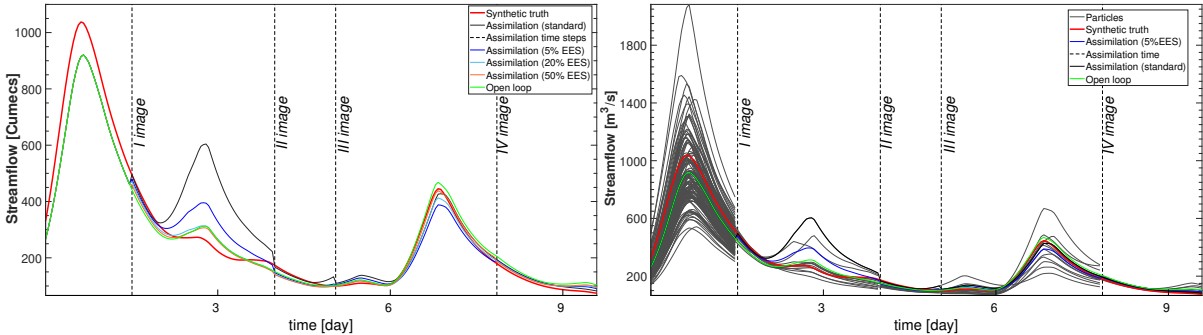

**Figure 11.** Streamflow time series at Bewdley. Left: Assimilation runs with an EES of 5%(blue), 20% (cyan) and 50% (orange), OL (green), standard assimilation (black). Right: particles carrying significant weight after the assimilation at 5% EES (grey). Dashed lines correspond to the assimilation times.

largest weight. All the 128 particles are equally weighted until the first observation is assimilated. After the first assimilation the number of important particles decreases. At the second assimilation time step, weights are recomputed and the new *important* particles are selected again and so on. The assimilation of the PFMs improves the predictions of water levels and streamflow at specific points of the river Severn, as in Bewdley and in Saxons Lode (Figure 10, 11), for the majority of the assimilation time steps in both underprediction and overprediction cases. The standard method and similarly the 5 % EES assimilation method

are the most accurate in forecasting the values of water levels and streamflows.

     The improvements due to the assimilation persist for a long time: up to 27 hours after the first assimilation predictions are still close to the synthetic *truth*. The local results of water levels suggest that the inaccuracy of the global RMSE values in time is likely due to the evaluation over the entire flood domain.





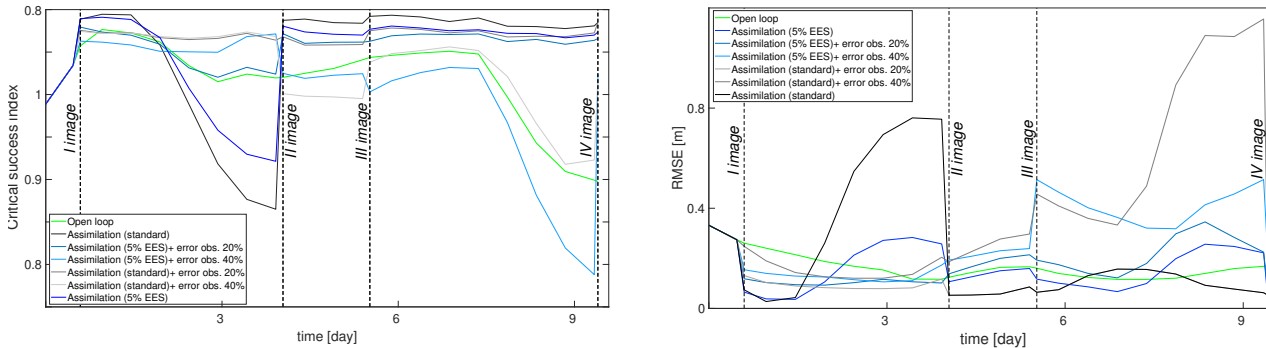

**Figure 12.** CSI values (on the left) and RMSE (on the rigth) after the standard assimilation of SAR observations free of errors (black), with 20% of errors (grey), with 40% of errors (ligth grey) and after the 5% EES assimilation free of errors (blue), with 20% of errors (ligth blue), with 40% of errors (cyan).

### 4.4 Errors in the observations

In the previous section, speckle uncertainty affecting SAR observations were considered. However, in reality, SAR observations are also susceptible to errors due to the misclassification of wet/dry pixels caused by features on the ground as already mentioned in the introduction. Therefore, a bias is added to the synthetic SAR observation as described in the methodology to investigate the impact on the DA assimilation framework. Figure 12 shows the RMSE and the CSI obtained at different assimilation time steps. The best performing assimilation methods (i.e. standard and 5% EES) with no bias in the observations

are compared with the ones where bias is introduced. With the misclassification of 20% of the pixels, the assimilation still has beneficial effects: the CSI increases at each assimilation time step with respect to the OL. The RMSE values also tend to be satisfactory after each assimilation. With an increase in the error of 40% the performances of the DA framework start decreasing. The assimilation of the first image still has a positive effect on the predictions. In fact, CSI and RMSE are improved with respect to the OL even through the improvements are not as significant as in the previous cases. The explanation is arguably

to be found in the high number of flood pixels. It is large enough to counterbalance the misclassified pixels in the SAR image. Performances decrease with the assimilation of the remaining SAR observations when the number of flood pixels is reduced by half.

### 5 Conclusions

Satellite images provide valuable information about flood extent that can complement or substitute in situ measurements.

The fact that several space agencies provide free access to high resolution satellite Earth Observation data paves the way for improving Earth Observation-based flood forecasting and reanalyses worldwide. This study represents a follow-up of the previous real case study from Hostache et al. (2018) with the objective to further proceed in the evaluation of the proposed DA framework once the assumptions are effectively satisfied. This study has been set up in a controlled environment using a





synthetically generated data-set in order to make sure that the rainfall and SAR observations are the only source of uncertainty.
A common issue in Particle filters is degeneracy: the ensemble could collapse after the assimilation because higher probabilities are assigned to a limited number of particles. Therefore, an enhanced coefficient, based on the desired effective ensemble size after the assimilation, has been tested. The tempering coefficient inflates the posterior probability to reduce degeneracy, reduces the peak of the likelihood and moves iteratively and smoothly from the prior to the posterior probability density function. A sensitivity analysis to the tempering coefficient of the Sequential importance sampling (SIS) filter has been proposed. In
addition, we investigated the impact of bias in the observations (i.e. errors in the SAR derived flood probabilistic maps due to dry water-look alike pixels or emerging objects) on the assimilation. Indeed, the main issue of using SAR observations in flood forecasting models is the difficulty of detecting flooded area for specific cases (e.g. urban or vegetated areas). At first, following the study from Hostache et al. (2018) only speckle uncertainty of the SAR image is taken into account in the Probabilistic flood maps. In a second step, a bias to reproduce misclassified pixels is introduced in the synthetic SAR observations.
The following key conclusions can be drawn from our experiments:

1. The best performing method is the standard method (i.e. SIS). Importance weights are assigned to a limited number of particles that better agree with the observations. At the time of the assimilation, results tend to be very accurate: the forecasts move close to the synthetic truth. The main weakness of the standard filter is to significantly suffer from degeneracy.

2. The 5% effective ensemble size assimilation (meaning that only the 5% of the ensemble will have a not-negligible weight after the assimilation) is slightly less accurate at the time of the assimilation but it has the advantage of reducing the degeneracy problem. Even though larger effective ensemble size prevents degeneracy results are less accurate and performances of the predictions are degraded.

3. The persistence in time of the improvements depends on the rapidity with which hydrological conditions change. A
frequent image acquisition could help in keeping model predictions on track especially when the dynamic of the system is varying fast and the persistence in time of the assimilation is not long enough.

4. Our study further shows that it is important to characterize and mask out errors in the SAR observations. An increasing number of wrongly classified pixels leads to a substantial reduction of the performances of the DA framework. In our case study, the improvement of model simulation (water levels and streamflow) and of performances (CSI and RMSE)
with the assimilation is only possible when the errors in the observation is not larger than 20% of pixels in the SAR image.

The results confirm the validity of the DA framework when the hypothesis of the rainfall as main source of uncertainty is verified from which it could be inferred that the wrong results, in the previous real case study by Hostache et al. (2018), at some assimilation time steps may be eventually explained by additional sources of uncertainties not taken into account.
Using flood probabilistic maps or backscatter values can provide more observations to be assimilated compared to a method that only derives the flood edge from satellite observations without suffering from the problems of the simple flood-edge



as reported in Cooper et al. (2019). The nearly-direct use of the SAR information has the additional advantage of a faster processing from the acquisition to the assimilation of the SAR image being therefore suitable for operational needs.

In our experiments, the improvements of model forecasts of water level and streamflow are high at the assimilation time step but they start decreasing after some hours (for example 27 hours after the first assimilation model results perform worse then the open loop) deviating from the synthetic truth. The update of a state variable of the forecasting model could increase the persistence in time of the improvements. In fact, in our study none of the model state variables is updated as only the particle weights are computed, based on the SAR observations and on the predicted flood extent maps, and used to calculate the expectation of water levels and streamflow. In previous studies [Andreadis et al. (2007), Matgen et al. (2010), Cooper et al. (2018)], it has been noted that water level updating determines an improvement in forecasts not as persistent in time as the inflow updating. For instance, one of the conclusions from the study by Matgen et al. (2010) was that updating the fluxes at the upstream boundary conditions, rather than the water levels, is more effective because of the high uncertainty of the inflow due to the poorly known rainfall distribution over the catchment. Therefore, as a future perspective, we foresee updating the state level reservoir of the hydrological model because it might have a positive impact on the long-term runoff simulations and consequently on the persistence in time of DA benefits.

The tempering coefficient as it has been used in this study could lead to biased results because it tends to down-weight the observations by increasing their errors. As described in Neal (1996) and in van Leeuwen et al. (2019), the tempering procedure consists of several steps, but in this study the tempering coefficient is applied only to flatten the likelihood, which maybe explains why data assimilation performs better when the effective ensemble size (the number of particle not negligible after the assimilation) is small. As already mentioned, the present study has the aim of assessing and validating the method proposed by Hostache et al. (2018) in a synthetic environment. Therefore, except for some minor modifications, the method here used is the same as proposed in the previous study. In a future study it is envisaged that to avoid degeneracy and keep a larger effective ensemble size, the full tempering scheme will be applied.

*Acknowledgements.* The research reported herein was funded by the National Research fund of Luxembourg through the HyDRO-CSI projects. Funding from the Austrian Science Funds as part of the Vienna Doctoral Programme on Water Resources System (DK W1219-N22) is acknowledged. Peter Jan van Leeuwen thanks the European Research Council (ERC) for funding of the CUNDA ERC 694509 project under the European Unions Horizon 2020 research and innovation programme. Nancy Nichols was funded in part by the UK Natural Environmental Research Council (NERC) National Centre for Earth Observation (NCEO).

The Lisflood-FP model can be freely downloaded at http://www.bristol.ac.uk/geography/research/hydrology/models/lisflood. The river cross-section data, the digital elevation model, and the gauging station water level, streamflow, and rating curve data are freely available upon request from the Environment Agency (enquiries@environmentagency.gov.uk). The ERA-5 data set is freely available at http://apps.ecmwf.int/datasets/data/interim-full-daily/levtype=sfc.



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
