# Peer review of "Assimilation of probabilistic flood maps from SAR data into a hydrologic-hydraulic forecasting model: a proof of concept."

_Hydrology and Earth System Sciences, 2020_

## Short Comment (SC1) · 5 Nov 2020

This review was prepared as part of graduate program course work at Wageningen University and has been produced under supervision of dr Ryan Teuling. The review has been posted because of its potential usefulness to the authors and editor. Although it has the format of a regular review as was requested by the course, this review was not solicited by the journal, and should be seen as a regular comment, we leave it up to the author's and editor which points will be addressed.

Floods represent one of the major natural disasters with a global annual average loss of US $104 billion, which emphasizes the need for reliable and cost-effective flood forecasting models. In this manuscript the authors aim to understand the main strengths and limitations of a previously proposed data assimilation framework in a fully controlled environment, in order to improve the quality of flood forecasting models. To do so, they performed synthetic twin experiments; At first, following the study from Hostache et al. (2018) only speckle uncertainty of the SAR image has been taken into account in the Probabilistic flood maps. In a second step, a bias to reproduce misclassified pixels is introduced in the synthetic SAR observations. The experimental results show that the assimilation of SAR probabilistic flood maps significantly improves the predictions of streamflow and water elevation, thereby confirming the effectiveness of the data assimilation framework.

This study is a follow-up of the previous real case study from Hostache et al., (2018). Major issues found in this study have led to this follow-up study, where two major things have been carried out differently. Firstly, Hostache et al. (2018) used a variant of the Particle Filter with Sequential Importance Sampling (SIS), to assimilate probabilistic flood maps (PFMs) derived from SAR data into a coupled hydrologic-hydraulic model with the assumption that the rainfall is the main source of uncertainty. This resulted in a reduction of the forecast errors; however, the improvements were not systematic: for some cases the updated hydraulic output deviates from the observations. The reason for such outliers could be the assumption that rainfall represents the dominating source of uncertainty together with satellite observation errors, excluding other possible sources of uncertainty in the model system. In this study the authors have decided to carry out a similar experiment but this time in a controlled environment so that rainfall is actually the only source of uncertainty.

Secondly, Hostache et al. (2018) highlighted that degeneracy may be a major issue of PFs. To overcome issue Hostache et al. (2018) used a site-dependent tempering coefficient which inflates the posterior probability. In this study, the authors adopted an enhanced tempering coefficient. The latter is a function of the desired effective ensemble size after the assimilation. The adapted method is compared to the standard

method where only one particle is left after the assimilation.

The manuscript is of societal significance, as floods represent one of the major global natural disasters. Therefore, the importance of this topic is to develop a reliable and cost-effective flood forecasting model, however, it is questionable whether this is achieved. In general, the improvement of model predictions is critical in reducing future material and immaterial damage caused by flooding. However, I found the relative contribution of this study in the improvement of flood model predictions unclear. It is indistinct how significantly this study contributes to improvements in global flood predictions. Whether the findings of this research are useful in other catchment areas over the globe is not defined.

Overall, I found the methodology and the results of this study worked out well. The stepwise approach is clear, and the representation of the results is very interesting to read. In the contrary, I found the introduction quite long, and very detailed. In my opinion, it difficult to find out what the broadly interesting knowledge gaps are. I suggest the introduction to be focused on the importance and contribution of this study to the appliance of flood model prediction on a global scale, and how that is achieved. Overall, this study fits well with the scope of the journal and can be published when several relatively minor issues are addressed. Below I provide more detailed comments.

Firstly, the introduction is very detailed on background information, but lacking in the focus of this specific research. The background information consists of a description of Data Assimilation (DA) and Synthetic Aperture Radar (SAR) images which is an essential part of the study. Different assimilation methods are discussed in detail, but it is not clear which and why this method is expected to be used. The detailed treatment of all the background information causes an unclear overview of what the study actually is about, as several DA methods are mentioned (KF, 4DVar, PF, EnKF (line 42-58). The introduction does not lead to a specific research question or clear objective. The objective is mentioned twice (line 74 and line 91), but the choice of words is different, which causes confusion. The objective is mentioned, but a briefly description on how

the objective is achieved is missing. The referencing of previous studies is used to discuss this, but it is unclear which references are really used for the methodology. Now, only the last sentence of the introduction is stating that 'a sensitivity analysis of the DA framework with respect to the tempering coefficient is conducted', which is rather vague.

For this major argument I would recommend being more specific with referring to previous studies, and to have a critical look on the broad background information. Especially, some of the referred studies concerning KF, 4DVar and PF seem unnecessary to me. Try to aim for a narrowing of the introduction, so that the introduction leads to the objective and research questions of this study. I would remove the objective mentioned in line 74, as it is to subtle and not agree with the objective stated in line 91.

The second argument concerns the conclusions of this study. Overall, I am very content with the conclusions. The conclusions of this research focus on the specific study area of the River Severn (UK), which is logically in line with the objective of this study. In line 376 it is stated that the main issue of using SAR observations in flood forecasting models is the difficulty of detecting flooded areas for specific cases, such as urban or vegetated areas. For now, this study only seems to be applicable for this specific study area, but I wonder if that is really the case. The societal significance of this study would be large if it contributes to a global improvement of flood modelling. I would recommend discussing the use of the findings of this study on a global scale. As a reader I would like to know how these results improve the appliance of Sar observations for different types of land use, or if a significant error increase is expected when the analysis is performed for different types of landscapes or land uses.

Thirdly, I did not really understand what is described in paragraph 2 of the methodology (line 139-144). In a previous study by Giustarini et al. (2016) the prior probabilities were proposed to be 0.5 as default value. The new methodology will probably lead to more area specific, but this is not confirmed by a reference. I am wondering if this method has been used before, or how the authors came up with this approach. The

difference between the 0.5 default value defined by Giustatrini et al. (2016) and the derivation from the true binary map in this study, could lead to significant differences in the chances of pixels being flooded or non-flooded. The improvement of the calculations by these methods would really confirm if it is valid, yes or no. To be short, I agree with the use of true binary maps (from true rainfall) to make a better estimation of the probability for each pixel of a SAR image, as true data is directly used to validate the SAR observations. However, I would recommend giving further details about the reasoning for choosing this methodology. There is no clear reason for using a more complicated method instead of the method by Giustarini et al. (2016). Below, some minor arguments and/or issues that I found in the manuscript have been described.

Minor argument 1: In line 68-69 it is mentioned that there could be other possible sources of uncertainty in the model system. I suggest some examples, as for now it is unclear in what direction these other sources uncertainties have to be found.

Minor argument 2: The spilling of water into the floodplain is modelled with a 2D diffusion wave scheme neglecting the convective acceleration (line 114). I would like to a reference or reasoning for the neglection of the convective acceleration. Even though it is logical, an assumption has been made about whether this term can be neglected, yes or no.

Minor argument 3: In line 118 the true meteorological is defined as temperature and rainfall. However, it is mentioned that the true rainfall data is used in the hydrological and hydraulic model. It is therefore unclear why temperature is taken into account in the true meteorological model. I would recommend giving a clarification on this by indicating how temperature is used in the model or remove it if it has not been used at all.

Minor argument 4: In this study, the alpha value is based on the desired effective ensemble size (EES) (line 222). It is unclear if this method has been used in previous studies and what the expected outcome of the use of this EES would be. I recommend adding some detail about this in the method section as it is a major change in comparison to the previous study by Hostache et al. (2018).

List of minor issues Introduction Minor issue 1: Line 68-69: It is mentioned that other sources of uncertainties could influence the model system. I think it is interesting to mention what these other uncertainties are.

Methods Minor issue 2: Line 116: "No later inflow in…". I assume this is incorrect. Shouldn't this be 'latent inflow'?

Minor issue 3: Line 216: "Since ïĄą and weights have values are lower than one", missing "that" before "are".

Results Minor issue 4: Line 274-276: It is unclear if this is correct. The revisit time is around 3-4 days, which means 2 satellite images per week. I do not understand how this results in four assimilated synthetic observations in a period of 10 days. 2 images per week $\approx$ 3 images per 10 days?

Minor issue 5: Fig 3 & 4: I have not read what the pixel size of the SAR observation is. I think it is important to mention the pixel size of the SAR observations by the Sentinal-1 satellite.

Minor issue 6: Fig 6: Labels incorrect. In the graph of streamflow time series (left) the assimilation of image I is indicated four times, while in the image on the right the labels assimilation of image I, II, III, IV are given.

Minor issue 7: Line 301 & Fig 7: "higher than the orange ones…". As the comparison between over detection (red) and under detection (black) is probably meant, "orange" should be "black".

---

## Author Comment (AC1) · 6 Dec 2020

Dear Arnoud Goossen, We would like first to thank you for the useful and detailed comments you have provided. We will take your suggestions into account in the revised version of the manuscript in case it will be accepted for revisions. We would like to seize the opportunity to clarify some aspects that you have pointed out in your review hereafter:

1) You pointed out that the introduction is long and does not focus enough on the specific research gaps and that too much room is given to the different Data Assimilation (DA) methods. In the intro-duction we explained that the number of studies aiming at

assimilating flood extent maps into flood forecasting models is small when compared to the ones using water levels derived from EO data. Among the studies assimilating flood extent, only few use a Particle Filter-based approach like the one introduced by Hostache et al. in 2018. In our opinion different assimilation methods are only briefly discussed in the introduction and no details are provided. We would argue that a brief de-scription of the main characteristics is useful being the selection of the most appropriate filter one of the open questions in this field of research. However, we will make sure that in the revised version of the introduction all unnecessary details will be removed.

2) Concerning the objectives, it is true that they are defined twice in a slightly different way and we agree that this may indeed create some confusion. The paper will be modified accordingly. We would like to clarify that the main objective of this study is to further assess the main strengths and limita-tions of a previously proposed DA framework by applying the method in a fully controlled environ-ment. A secondary objective is to propose a new framework for evaluating the performance of dif-ferent DA approaches for assimilating Synthetic Aperture radar-derived flood probability maps into a hydraulic model.

3) With respect to the apparent lack of clarity concerning the contribution of this paper to the improvement of flood model predictions, we will add more explanations to the revised version of the manuscript. We argue that the method used in the paper has the potential to support EO- based modelling in sufficiently large floodplains where flood inundations remain visible to satellite sensor over a sufficiently long period of time. Indeed, these two main constraints must be satisfied to enable the application of the proposed framework and to make use of the analysis carried out in this manuscript. Moreover, it is worth noting that the proposed DA framework can be applied to a variety of flood inundation forecasting chains. The main reason for this is that it does not require any updating of the state variables and parameters of the model.

4) In Giustarini et al. (2016), the prior probability is assumed to be 0.5 since no infor-
mation on the prior can be obtained. In this paper, however, because of the synthetic nature of the experiment, the prior is known as it can be derived from the true binary flood extent maps. Moreover, we have also carried out the experiment with a default value of the prior (0.5) and found that its value has no effect on the results of the experiment, as explained in the lines 279 -283 of our paper.

- Minor argument 1: "It is mentioned that there could be other possible sources of uncertainty, I sug-gest some examples". It is true that other sources of uncertainty: "…input data, model parameters, in-itial conditions and model structure represent sources of uncertainty that affect the reliability and ac-curacy of flood forecasts" were only mentioned in the abstract (lines 2-3). We will add such exam-ples to another section of the paper as well.

- Minor argument 2: The neglection of the convective acceleration for the simulation of the spilling of the water in the floodplain is rather common in order to simplify the shallow water equation. The 2D solver that has been used in the floodplain is the acceleration solver (Bates et al., 2010; De Almeida et al., 2012) which neglects only the convective acceleration. In addition, the hydraulic model used here is based on the set-up defined in Melissa Wood et al. (2016) as mentioned in line 116.

- Minor argument 3: The temperature is considered to calculate the evapotranspiration term in the hydrological model.

- Minor argument 4: You have said that it is unclear if the $\alpha$ value based on the desired effective ensemble size has been used before. We would like to explain that the use of a tempering coefficient based on the effective ensemble size (EES) is commonly used in Particle Filters (van Leeuwen, 2019). The EES is used to obtain an idea of the number of particles that will have a not negligible weight after the assimilation.

- Minor issue 1: In line 68-69 it is mentioned that other sources of uncertainty could influence the model system, it is true that these sources are only listed in the abstract and in a future version of the paper we will take into account this comment.

- Minor issue 2: "no later inflow in . . ." is incorrect. This is a typo. Indeed, it should be written "lateral inflow".

- Minor issue 4: The revisit time is around 3- 4 days, which means 2 Sentinel-1 satellite images are acquired on average every week. The revisit time for a single orbit is 6 days but in our case study we are considering many orbits (ascending and descending) so the revisit time will be 3-4 days as shown in the enclosed image, which means 2 satellite images per week. The combination of orbits will give us an image with a higher frequency of acquisition compared to the single orbit. The acquisition dates used in the paper were actually derived from real acquisition dates of SENTINEL 1 over the area in a different year. When a third satellite of the constellation will become operational, the revisit time can be lowered even further.

- For the other minor issues reported we agree that those corrections should be made in the new version of the paper in case it will be accepted for revisions.
* * *
[Figure]

**Fig. 1.**

---

## Referee Comment (RC1) · Anonymous Referee #1 · 7 Dec 2020

The paper addresses a topic of immense community interest. The methodological design is sound, and the overall writing quality is quite good. However, this paper lacks scientific/conceptual contribution.

The main contribution of this paper is rather technical. I say this because the concept of assimilating remotely sensed flood maps into flood models is not new. While the authors nicely rationalized their limited focus (by clarifying that their goal is to assess previous DA frameworks and draw generic conclusions; see P4), I see a major conceptual issue which may put this paper in a "conflicting" position against the current state of science. See below.

The methodology presented in this paper is not applicable to the common practice of flood inundation modeling/forecasting. Specifically, regardless of DA technique (e.g., particle filter), effect of SAR observations cannot be fed back to streamflow and stage height unless the hydrology and hydrodynamic models are tightly coupled. Most of the large basin/continental-scale flood inundation forecasting frameworks rely on loosely coupled hydrology (A) and hydraulic (B) model components. In such a framework, there is only a one-way transfer of information from A to B using a relational data-model (Peckham et al., 2013). The VIC and LISFLOOD-FP coupling by Schumann et al. (2013), the VIC, Delft3D, and LISFLOOD-FP coupling (GLOFRIM framework) by Hoch et al. (2017), and the more recent SWAT and LISFLOOD-FP coupling by Rajib et al. (2020), all rely on loose coupling of models; as such, the approach presented in this paper (and the underlying math) cannot be generalized. Therefore, I strongly suggest adding a separate paragraph in the introduction highlighting this limitation (dear authors: please feel free to recycle the above texts and references when you revise your paper). Accordingly, I also recommend editing the title as "Assimilation of probabilistic flood maps from SAR data into a coupled hydrologic-hydraulic forecasting model: a proof of concept".

Peckham et al., 2013: https://doi.org/10.1016/j.cageo.2012.04.002; Schumann et al., 2013: https://doi.org/10.1002/wrcr.20521; Hoch et al., 2017: https://doi.org/10.5194/gmd-10-3913-2017; Rajib et al., 2020: https://doi.org/10.1016/j.jhydrol.2019.124406;

---

## Author Comment (AC2) · 23 Dec 2020

We would like to thank the Referee #1 for the careful reading of the manuscript and the valuable comments provided. We agree with the suggestion to change the manuscript title in order to add the word "coupled". Referee #1 argues that the concept of assimilating remotely sensed flood maps into flood models is not new. Regarding this statement, we would like to highlight that only few recent studies (a handful at maximum) have proposed methods to assimilate flood extents into flood inundation models, Lai et al. (2014) being the first one to our knowledge. We believe this shows that the technique is recent and there are still plenty of related research questions to answer

before this can be routinely applied. In particular, the method proposed by Hostache et al. (2018) is new and needs further investigation to better understand its current limitations and strengths and therefore assess its global applicability. Moreover, in addition to the validation of the approach in a controlled environment, our study demonstrates the importance of an enhanced tempering method to avoid the degeneracy and for a bias removal in the SAR observations prior to the assimilation. Referee #1 also raises the concern that SAR observations effect can be fed back to streamflow and stage height only if a tight coupling of the models is used and consequently that the proposed approach is not globally applicable considering that most of the current flood inundation forecasting frameworks rely on loosely coupled models. We would like first to clarify that our implementation of a particle filter relies on a sequential importance sampling (SIS) that does not include a variable update step. Indeed, only the particle weights are updated based on the observation and then used to compute the expectation (weighted mean) of the augmented state vector (including hydraulic state variables of water depth, plus flood extents and boundary conditions). In our forecasting system, the hydrological model (SUPERFLEX) and the hydraulic model (LISFLOOD-FD) are loosely coupled similarly to the other studies reported in the review. Indeed, the streamflow simulated with SUPERFLEX feeds LISFLOOD-FP as upstream boundary conditions to simulate flood inundation extents. This also implies that the domains of the two models are not overlapping, but they are only connected at the upstream boundary condition of the hydraulic model. Moreover, we acknowledge that the flood wave entering the hydraulic model via the upstream boundary conditions needs some time to propagate across the hydraulic model domain. Conse-quently, the observed flood extent is closely linked to past time steps of the boundary conditions rather than the current (i.e. at the assimilation time) boundary conditions value. Therefore, when computing the weights (analysis step) based on the observed flood extent, the best performing particles are the ones having past forcing (upstream boundary conditions of the hydraulic model) values closer to the truth. It could be argued that the particles that were better in the past should be equally better at the assimilation time and for

subsequent time steps. This is confirmed by figure 10 and 11 were the expectations (i.e. analysis weighted mean) of the stream water depth (at Saxon's Lode, within the hydraulic model domain) and the upstream discharge (at Bewdley, upstream boundary condition of the hydraulic model) are closer to the truth than the open loop at every assimilation step. Of course, we recognize that spurious correlations may occur between SAR observations and the model variables due to limited ensemble size. Enlarging ensemble size could be necessary if this occurs. Indeed, the objective of a future study will be to address this issue.

We will add this detailed explanation on the validity of the approach based on a loose coupling in the results and discussion part of the revised version of the article, if accepted. As a conclusion, based on the above elements, we believe that our approach based on usual state augmentation is valid, regardless of the type of model coupling, and applicable to many different forecasting systems.

Please also note the supplement to this comment:
https://hess.copernicus.org/preprints/hess-2020-403/hess-2020-403-AC2-supplement.pdf

**Supplement:**

Dear Editor,

Please consider that we are going to provide a revised version of the manuscript to add the aspects raised with the first review in the beginning of the next month.

Thanks for your time and your consideration.

Kind Regards

---

## Referee Comment (RC2) · Anonymous Referee #2 · 11 Jan 2021

This is a highly technical manuscript focused on assimilation of many different data sources using multiple techniques to predict flood extent and depth. I think this is an interesting study, but I overall I think it needs major improvements before it can be published. The science is sound and interesting, but the manuscript could be clarified and revised throughout to make this easier for the reader to understand. I summarize my major comments and minor comments below.

Major points: My main recommendation to the authors is to clearly clarify the contribution of this study to the literature. The manuscript incorporates many technical methodological assessments, but it is not always clear why these assessments are be-

[Figure]

ing conducted, and what they help us learn about flood modeling. The authors should clearly state their contributions in the introduction, and clarify in a discussion section how their findings advance those conducted by other studies.

The introduction should be revised and reorganized. At current, the introduction is very technical, and describes a lot of the existing literature. However, I had a hard time following the common threads and major points being made across the arc of the introduction. Many individual references are described, but aren't necessarily connected to the bigger picture of flood modeling. More synthesis is needed across these references and paragraphs to highlight the major knowledge gaps. Furthermore, I'd recommend shortening the introduction. Finally, the introduction section normally concludes with a statement about the novelty of the study, the scope, and the objectives. These are instead first introduced on line 75, then again later in the introduction. I'd recommend consolidating these statements into a coherent paragraph at the end of the introduction.

At the end of the introduction, I am left unsure of the scope and objectives of the manuscript (for instance, nothing about SAR or flooding is mentioned). These three concluding sentences could benefit from more specifics as to what will be tested and explored in this particular article. Specifics, such as types of model used, data resolution, etc could be specified here, to more clearly articulate to the reader the framing of your particular work.

The methods section is very detailed (which I appreciate). Yet, I had a hard time understanding the major comparisons to be made in the results/discussion section. Could you more clearly summarize these and why you are comparing these methods at the start of this section? The workflow is helpful, but with the number of methods and acronyms, I had a hard time following this.

The Study Area section comes after the methods section – this was a little confusing to me, because the nuances of this are discussed in the methods section. Is it worth switching the order of these?

This may be my own personal preference, but I've been taught a paragraph should be 3 or more sentences. There are many cases where there are paragraphs of one or two sentences (e.g., line 285). Please ensure that all paragraphs are 3+ sentences, and ensure that these are appropriately combined throughout the text.

I would recommend relabeling sub-sections within the results to separate out the different comparisons and techniques you are making – organizing these headings would help me connect what you do to your methods section. For instance, I had a hard time connecting these results to the stated objective of detecting uncertainty in precipitation, and then to the conclusions section. It could also help to start each sub-section by describing what methods/approaches you are testing and why, given there are many comparisons.

At current, the conclusions section is quite long and there is no discussion section. This may be a personal preference, but I would recommend shortening the conclusions section, and moving much of what is in there now to a discussion section. Within this discussion section, the main piece I don't see is a discussion of the limitations of this approach – for instance, you consider one event – is there a reason to think that this approach is transferable? Why or why not? In what scenarios is this approach most useful (ie., at what scale)? Given rainfall is the main source of uncertainty, what does this mean for future work? Can this work improve forecasting?

Minor points: Line 42 – "used" is repeated Line 48-49 – I had trouble understanding this sentence – could you rephrase? It was not clear to me what 'the latter' referred to Line 42 – I am missing the connection from this paragraph to the next - why would one want to use a KP, 4DVar, or PF technique for assimilation of flood information? Can you connect these thoughts to the previous sentence? Line 52 – is there a reason to have a whole paragraph focused on this particular article? Is it most similar to what is done in this study? Do you improve on their work? If not, I'd recommend shortening the description of this article. Lines 76 – 90 – this is very detailed, to the point where I am unsure if this is helpful in the introduction. Would you be able to shorten this section

and distill a few key messages? Could this be moved to the methods section? Line 178: "supposed to be uniform" – do you mean assumed to be uniform? Sampled as uniform? Please clarify. Section 2.3 – please weave the equations into the text, instead of listing them after the text here Section 2.4 – please do a thorough read to ensure that all variables in the contained equations are clearly defined in this section Lines 228 – 232 – this reads as 'results' – should this be moved to the results section? Section 3.0 – please capitalize 'area' Line 276: The plots in this section show four time points – why did you select these time points? Please introduce the time points in this section. Line 285: You show a sub-section of the result area multiple times – please introduce this area and why you selected it in the text. Also – are you computing results for just this section of the river or the entire watershed? I wasn't sure from the methods and study area section. Please clarify. Line 274 – 284 – should this be in methods? Line 279 – 281 – what is the significance of this? Could you explain more why you mention this here? This again seems like 'methods' – should this be moved to the methods section, or is it a 'result' of your investigation? Line 284 – Figure 3 and Figure 5 are mentioned – figures should be listed in order. Figure 4 is not cited in the text. Should this be removed or moved to Supporting Information? Line 315 – Please do not start a sentence with a number Line 387 – 399 – could you rephrase this sentence? I don't understand what it is saying.

Figure 2: Could you highlight on this figure the places you select for Figure 3 and Figure 4? Figure 3 & 4: The legend is hard to see, and there is no label of what 'value' is being shown (and its associated units). Figure 3 & 4: What are the four assimilation time steps? Please label these figures as (a)-(d) or on the figure to indicate this. Figure 3 & 4: Should these be combined to enable comparison? It is not entirely clear from the results text what these images show and how these connect to the workflow.

Table 1 & Table 2: Please direct readers to Figure 6 in the captions for these.

[Figure]

403, 2020.

---

## Author Comment (AC4) · 8 Feb 2021

**Reply to the review nr.2**

February 2021

We would like to thank referee for the careful reading and the very useful comments. Below we address the referee comments and explain how the manuscript will be updated. Comments of the referee are in italic and bold font.

***This is a highly technical manuscript focused on assimilation of many different data sources using multiple techniques to predict flood extent and depth. I think this is an interesting study, but I overall I think it needs major improvements before it can be published. The science is sound and interesting, but the manuscript could be clarified and re-vised throughout to make this easier for the reader to understand. I summarize my major comments and minor comments below.***

We will clarify the manuscript as suggested by the referee.

**1 INTRODUCTION**

1. ***My main recommendation to the authors is to clearly clarify the contribution of this study to the literature. The manuscript incorporates many technical methodological assessments, but it is not always clear why these as-***

*sessments are being conducted, and what they help us learn about flood modelling. The authors should clearly state their contributions in the introduction and clarify in a discussion section how their findings advance those conducted by other studies.*

This study is a follow up of Hostache et al. (2018) who applied a Particle Filter (PF) assimilation to a real case study at the River Severn which however, resulted in an overestimation of streamflow in some cases. In this study we identify the reasons for this overestimation using synthetic experiments in order to be able to exactly control the assumptions, and more generally assess the strengths and limitations of the method. Additionally, we improve their method to overcome degeneracy issue, and evaluate the effect of pixel misclassification in SAR observations on the DA performance. We will now more clearly state the contributions in the introduction and add a discussion section to spell out the advances of the paper. Further explanations are provided later in this document (paragraphs 4.1 and 4.2).

2. *The introduction should be revised and reorganized. At current, the introduction is very technical, and describes a lot of the existing literature. However, I had a hard time following the common threads and major points being made across the arc of the introduction. Many individual references are described but aren't necessarily connected to the bigger picture of flood modelling.*

We thank referee for these relevant remarks and the manuscript will be updated accordingly.

(a) In the first part a brief introduction to DA of satellite observations is made with the following key points: the importance of flood forecasting (19 -24 lines of the manuscript), the advantages of using satellite observation and adequate DA techniques (25 -32 lines of the manuscript), the SAR image

acquisition characteristics and the information on flood extent that can be extracted and assimilated into a model (33 -36 lines of the manuscript).

(b) In the second part of the introduction we focus more specifically on the existing methods for assimilating flood extent maps into forecasting models. Most of them (36 -41 lines of the manuscript) transform the flood extent information into state variables of the model such as water levels or discharge [e.g. García-Pintado et al. (2015), Matgen et al. (2010), Revilla-Romero et al. (2016), Giustarini et al. (2011), Hostache et al. (2010)], other do not require this transformation into a model state variable and thus allow directly assimilating backscatter or probabilistic flood maps [Lai et al. (2014), Revilla-Romero et al. (2016), Cooper et al. (2018), Cooper et al. (2019), Hostache et al. (2018)] (lines 41-57of the manuscript). These direct assimilation techniques have been proposed only very recently and there are still numerous open research questions to answer before they can be applied routinely.

(c) In the third part of the introduction, we indeed focus on the study of Hostache et al. (2018) that requires further investigations to enable a better understanding of current limitations and strengths. Some main differences between the Particle filter and other assimilation techniques are defined (lines 58-61 lines of the manuscript) to explain the choice of the Particle Filter in Hostache et al. (2018) and in the current study. More details (lines 68 -75 of the manuscript) are provided on the Hostache et al. (2018) method and on the assumptions made on the rainfall as the main source of uncertainty. We think that such information is relevant in the introduction as it introduces the reasons that have led the authors of this manuscript to conduct a synthetic experiment. The remaining lines define two main issues that are being addressed in the manuscript: lines 76 - 80 introduce the degeneracy problem and the different methods adopted in the paper (standard and adapted methods), lines 81-90 describe the issue of

misclassification of SAR pixels.

The introduction will be shortened in accordance with the referee's recommendations. Less detail on the existing literature (from 42 – 57 lines of the manuscript) will be given. Moreover, we agree that the paragraph on lines 58-61 could create some confusion as it is not completely aligned with the rest of the introduction and will therefore be removed. With these adjustments, we believe that the reader can better follow the main reasoning and the major points of the introduction.

3. ***More synthesis is needed across these references and paragraphs to highlight the major knowledge gaps. Furthermore, I'd recommend shortening the introduction.***
In the literature, there are very few studies directly assimilating SAR-derived flood inundation information into a forecasting model. The most commonly used method is the transformation of SAR-derived information into a state variable of the model, namely into water levels. We argue that the methodology here presented is very novel and that the method proposed by Hostache et al. (2018) needs further investigation to better understand its current limitations and strengths and to therefore assess its applicability at large scale. We agree that the introduction should be shortened. Revilla-Romero et al. (2016), Lai et al. (2014), and Cooper et al. (2019) are some examples of studies where the information derived from SAR data is not transformed into a state variable of the model before being assimilated. To make the introduction more focused, we will condense this part and remove unnecessary details about these methods. From line 58 up to line 61, some differences between PFs and other assimilation techniques are mentioned. This part will be removed.

4. ***Finally, the introduction section normally concludes with a statement about the novelty of the study, the scope, and the objectives. These are instead***

*first introduced on line 75, then again later in the introduction. I'd recommend consolidating these statements into a coherent paragraph at the end of the introduction.*

We fully agree that objectives are defined with two separate statements. The paper will be modified accordingly with a "coherent paragraph" at the end of the introduction pointing out these objectives:

(a) The main objective of the manuscript is to evaluate the strengths and limitations of the DA framework proposed by Hostache et al. (2018) with a synthetic experiment where rainfall, together with SAR observations, are the only sources of uncertainty.

(b) The second objective is to further develop this DA framework for combating degeneracy more efficiently.

(c) The last objective is to evaluate the effects of misclassification in the SAR-derived observations on the performances of the PF.

5. *At the end of the introduction, I am left unsure of the scope and objectives of the manuscript (for instance, nothing about SAR or flooding is mentioned). These three concluding sentences could benefit from more specifics as to what will be tested and explored in this particular article. Specifics, such as types of model used, data resolution, etc could be specified here, to more clearly articulate to the reader the framing of your particular work.*

We will add a brief statement at the end of the introduction section, where we state some specifics as pointed out by the referee, but more details are given in the methods section: "the proposed forecasting system consists in a loose coupling of a hydrological (SUPERFLEX) and a hydraulic model (LISFLOOD-FP). The meteorological data are derived from the ERA-5 archive with a spatial resolution of 25 km and a temporal resolution of 1 hour. The SAR data are synthetically generated with a resolution pixel spacing of 75 m. Experiments are carried

out to evaluate the standard and the enhanced version of PF, and the effects of pixels misclassification of SAR observations on the DA."

**2 METHODS**

1. ***The methods section is very detailed (which I appreciate). Yet, I had a hard time understanding the major comparisons to be made in the results/discussion section Could you more clearly summarize these and why you are comparing these methods at the start of this section?***
   In the section 2 (from line 98) we will add a paragraph summarizing the different experiments:

   (a) The standard filter where degeneracy occurs;

   (b) The adapted method where a tempering coefficient is used to avoid degeneracy. A sensitivity analysis of the tempering coefficient is realized. Different tempering coefficients based on the desired effective ensemble size after the assimilation (5%-10%-20% and 50%) are used;

   (c) The proposed methods are also applied with known errors in SAR image classification in order to evaluate and understand the impact of these errors on the DA.

2. ***The workflow is helpful, but with the number of methods and acronyms, I had a hard time following this.***
   The workflow, (figure 1) is the same for all these tests. The "assimilation" represented by a blue circle is the only element changing between the standard filter and the adapted filters. We will explain the acronyms in the flow chart caption to make it more easily readable.

3. ***The Study Area section comes after the methods section – this was a little confusing to me, because the nuances of this are discussed in the methods section. Is it worth switching the order of these?***

   The reason why we have put the study area after the methods section is due to the fact that in the methods section we give a more general overview of the methodology (not related to the study area) and to show that it is applicable to many cases, whereas the study area is more specific, closer to our particular situation. We will ensure that the method part is free from site-related discussion of information.

4. ***Please ensure that all paragraphs are 3+ sentences, and ensure that these are appropriately combined throughout the text.***

   We will pay attention to this and ensure that all paragraphs contain at least 3 sentences.

**3 RESULTS**

1. ***I would recommend relabelling sub-sections within the results to separate out the different comparisons and techniques you are making – organizing these headings would help me connect what you do to your methods section.***

   We will restructure the results section in order to match the structure of the methods section.

2. ***For instance, I had a hard time connecting these results to the stated objective of detecting uncertainty in precipitation, and then to the conclusions section. It could also help to start each sub-section by describing what methods/approaches you are testing and why, given there are many comparisons.***

As recommended by the referee, we will re-organize the results section and will also carefully explain at the beginning of each subsection of the results section what will be tested and compared, in order to make the section clearer.

**4  CONCLUSIONS**

1. ***This may be a personal preference, but I would recommend shortening the conclusions section, and moving much of what is in there now to a discussion section.***
   We agree with this suggestion and we will revise the paper accordingly (see paragraphs 4.2 of this document).

2. ***Within this discussion section, the main piece I don't see is a discussion of the limitations of this approach.***
   The limitations, that are currently reported in the conclusions section, will be moved to the discussion section:

   (a) "Even though larger effective ensemble size prevents degeneracy, results are at the same time less accurate and performances of the predictions are degraded (line 387 of the manuscript)" . . . which is maybe due the fact that. . .. "in this study the tempering coefficient is applied only to flatten the likelihood" and the full tempering scheme in not applied (line 416 of the manuscript).

   (b) The study shows the validity of the DA framework if the uncertainty derives only from the precipitation and the SAR observations. Additional sources of uncertainties could be possibly considered for some real cases: it could be the case of Hostache et al. 2018 (line 397-399 of the manuscript).
   In the discussion section, we will also follow the referee's suggestions and

add the following remarks:

(c) ***Is there a reason to think that this approach is transferable? Why or why not?***
The proposed DA framework can be applied to a variety of flood inundation forecasting chains since the model updating is carried out via a sequential importance sampling only (i.e. importance weights).

(d) ***In what scenarios is this approach more useful (i.e. at what scale)?***
We argue that the method used in the manuscript has the potential to support EO-based modelling in sufficiently large floodplains where flood inundations remain present over a sufficiently long time period to be detectable to satellite sensors given their revisit interval. Indeed, this constraint must be satisfied to enable the application of the proposed framework and to make use of the analysis carried out in this manuscript.

(e) ***Given that rainfall is the main source of uncertainty, what does this mean for future work?***
For those cases where rainfall represents the main source of uncertainty, for example in poorly gauged or ungauged catchments or in forecasting models, our study results indicate that the application of the approach described in the manuscript will lead to improved results of the model simulations. Some modifications of the DA framework are still required to fully overcome the issue of degeneracy. For those cases where the uncertainty of other sources is more relevant, these sources need to be taken into account explicitly. Possible ways to adapt and advance the proposed DA framework are currently under development (e.g. updating a state variable of the model, using an enhanced version of the adapted filter).

(f) ***Can this work improve forecasting?***
It is shown in the paper that the assimilation is beneficial as it reduces forecast errors not only at the assimilation time steps, but also for subsequent times steps. The persistence in time of these "improvements depends on the flashiness of the flood event, i.e. "the rapidity with which hydrological conditions change" (line 389 of the manuscript). More frequent image acquisitions could help in keeping model predictions on track, especially when the dynamics of the system are fast. In future studies, we will evaluate if additionally updating the state variable helps in obtaining longer-term positive impacts on the simulations.

**5  MINOR POINTS**

1. ***Line 42 – "used" is repeated Line 48-49 – I had trouble understanding this sentence – could you rephrase? It was not clear to me what 'the latter' referred to:***
   We will change this sentence as follows: "In the existing literature only few studies have used DA for assimilating flood extent maps into flood forecasting models [e.g. Lai et al. (2014), Revilla-Romero et al. (2016), Cooper et al. (2018), Cooper et al. (2019), Hostache et al. (2018)]. The reason is the difficulty of directly assimilating flood extent since this is not a state variable of the model. Consequently, in many assimilation studies the flood extent information is transformed into water level as this is a state variable of most hydraulic models."

2. ***Line 42 – I am missing the connection from this paragraph to the next - why would one want to use a KP, 4DVar, or PF technique for assimilation of flood information? Can you connect these thoughts to the previous sentence?:***
   These assimilation techniques are mentioned because they are used in the cited references [Lai et al. (2014), Revilla-Romero et al. (2016), Cooper et al. (2018), Cooper et al. (2019), Hostache et al. (2018)] (lines 43 -44 of the manuscript). The

paragraph could be restructured as follows: "In the existing literature, only few studies have used DA techniques, such as Kalman Filter (KF), Four-Dimensional Variational (4DVar) and Particle Filter (PF), for directly (without any transformation into a model variable) assimilating flood maps into flood forecasting models".

3. ***Line 52 – is there a reason to have a whole paragraph focused on this particular article? Is it most similar to what is done in this study? Do you improve on their work? If not, I'd recommend shortening the description of this article:***
The reason why we are mentioning the article is to give an example of studies where the information derived from the SAR image is not transformed into a state variable of the model before the assimilation. We want to highlight the fact that "the new observation operator performs well compared to the assimilation of flood-edge water elevation observations". In the context of our study, it means that using information derived from SAR without transforming it into a model variable is an interesting option. The advantage of this technique is its applicability for operational uses in near- real time assimilation. Moreover, we agree that this paragraph should be reduced in the revised version of the paper. Therefore, it will be changed as follows: "Cooper et al. (2019) have used an EnKF to update a 2D hydrodynamic model. In this study, the backscatter values are directly assimilated into the model. The study has shown that the SAR backscatter-based assimilation method performs well compared to the EO-derived water levels assimilation."

4. ***Lines 76 – 90 – this is very detailed, to the point where I am unsure if this is helpful in the introduction. Would you be able to shorten this section and distil a few key messages? Could this be moved to the methods section?:***
As suggested, we will shorten this paragraph and move some technical details to the methods section as follows: "Moreover, Hostache et al. (2018) highlighted that degeneracy may be a major issue of PFs: after the assimilation, the number

of particles with significant weight reduces significantly to few or one particle so that the ensemble loses statistical significance. To overcome this issue Hostache et al. (2018) used a site-dependent tempering coefficient which inflates the posterior probability. In our study, we adopt an enhanced tempering coefficient defined as a function of the desired effective ensemble size after the assimilation. Moreover, there could be errors in the detection of flooded areas in SAR images. Detecting and removing these errors represents one of the main scientific challenges of using SAR data for a systematic, fully automated, and large-scale flood monitoring. In Hostache et al. (2018), speckle errors are therefore taken into account, but no conclusions are given on the effect of misclassified pixels in the SAR observations. Consequently, in this synthetic experiment, misclassification errors are artificially added to the SAR-derived flood extent with the aim to assess the robustness of the proposed method with respect to this type or errors."

5. ***Line 178: "supposed to be uniform" – do you mean assumed to be uniform? Sampled as uniform? Please clarify:***
Yes, we meant "assumed to be uniform", meaning that each particle has the same weight before the assimilation. This will be corrected.

6. ***Section 2.3 – please weave the equations into the text, instead of listing them after the text here:***
We will do so.

7. ***Section 2.4 – please do a thorough read to ensure that all variables in the contained equations are clearly defined in this section:***
We will do so.

8. ***Lines 228 – 232 – this reads as 'results' – should this be moved to the results section?:***
Lines 231 - 232 will be rephrased and moved to the conclusions section: "This

methodology leads to slightly biased estimates because the observation are down-weighted."

9. ***Section 3.0 – please capitalize 'area':***
We will take this into account.

10. ***Line 276: The plots in this section show four time points –why did you select these time points? Please introduce the time points in this section:***
Over Europe, the current revisit time is around 3- 4 days, which means 2 Sentinel-1 satellite images are acquired on average every week. The revisit time for a single orbit is 6 days but in our case study we are considering many orbits (ascending and descending) so the revisit time will be 3-4 days as shown in the enclosed image. The acquisition dates used in the manuscript were derived from a realistic acquisition plan of SENTINEL-1 over the area. They correspond to: 22-07-2007 10:00, 24-07-2007 17:56, 25-07-2007 17:49, 28-07-07 07 09:00. Please note that with another satellite being added to the constellation and other satellite missions being considered the revisit time as well can be further shortened.

11. ***Line 285: You show a sub-section of the result area multiple times – please introduce this area and why you selected it in the text. Also – are you computing results for just this section of the river or the entire watershed? I wasn't sure from the methods and study area section. Please clarify:***
This sub-area represents the domain of the hydraulic model. We compute and compare results along the main River Severn within this sub-area of the watershed which represents the flood-prone zone. The hydrological model with which the boundary conditions are evaluated covers the contributing upstream catchment shown in figure 2 of the manuscript.

12. ***Line 274 – 284 – should this be in methods?:***
Yes, we agree and will move this paragraph in the method section.

13. ***279 – 281 – what is the significance of this? Could you explain more why you mention this here? This again seems like 'methods' – should this be moved to the methods section, or is it a 'result' of your investigation?:***
    Yes, this will be moved to the methods section. In Giustarini et al. (2016), the prior probability is assumed to be 0.5 since no information on the prior can be obtained. In this paper, however, because of the synthetic nature of the experiment, the prior is known as it can be derived from the true binary flood extent maps. We have carried out experiments with a default value of the prior (0.5) and with the estimated prior and found that its value has no significant effect on the results of the experiment, as explained in lines 279 -283 of our manuscript.

14. ***Line 284 – Figure 3 and Figure 5 are mentioned – figures should be listed in order. Figure 4 is not cited in the text. Should this be removed or moved to Supporting Information?:***
    This is a typo because it should be written as follows: "the PFMs are shown in figure 4 and the corresponding reliability plots in Figure 5". This will be corrected.

15. ***Line 315 – Please do not start a sentence with a number:***
    We will take this into account.

16. ***Line 387 – 399 – could you rephrase this sentence? I don't understand what it is saying:***

    (a) Although the use of a smaller tempering coefficient leads to a larger effective ensemble size (e.g. 50 % ) and helps avoiding degeneracy, the results are less accurate compared to the standard method or a 5% EES method.
    (b) The persistence in time of the beneficial effects of the assimilation varies according to the rapidity of variations of flood extent; a more frequent image acquisition could help in better keeping the predictions on track.
    (c) Our study further shows that it is important to characterize and mask errors in the SAR observations. A large number of misclassified pixels substantially

degrades DA performance. In our study, the improvement of model simulations (water levels and streamflow) and performances (CSI and RMSE) after the assimilation is still possible if the errors in the SAR observations are rather limited (not more than the 20% of the pixels). However, if the misclassification goes beyond 40% of the pixels, the assimilation has no effect or even degrades the model predictions.

(d) The results confirm the validity of the DA framework when the hypothesis of the rainfall as main source of uncertainty is verified. This confirms that the limitations identified in the previous real case study by Hostache et al. (2018) could be explained by additional sources of uncertainties that were not taken into account.

17. ***Figure 2: Could you highlight on this figure the places you select for Figure 3 and Figure 4? Figure 3 and 4: The legend is hard to see, and there is no label of what 'value' is being shown (and its associated units). Figure 3 and 4: What are the four assimilation time steps? Please label these figures as (a)-(d) or on the figure to indicate this. Figure 3 and 4: Should these be combined to enable comparison? It is not entirely clear from the results text what these images show and how these connect to the workflow. Table 1 and Table 2: Please direct readers to Figure 6 in the captions for these:***

The remaining corrections of the figures will be made according to the referee's suggestions.

---

## Author Response (AR1)

This document contains our answers to the referees' and editor's comments. We would like first to thank Referees #1 and #2, the student Arnoud Goossen and the editor for the careful reading of the manuscript and their relevant remarks and comments. In what follows, the referees' remarks are written in black while our answers are written in blue. Moreover, extracts from the revised manuscript are written in magenta.

Enclosed are two versions of our revised paper: one without *track changes* and one with *track changes* [the removed elements in the manuscript are  while the added elements are in blue].

**1. Student review**

- **GENERAL COMMENTS**

  The manuscript is of societal significance, as floods represent one of the major globalnatural disasters. Therefore, the importance of this topic is to develop a reliableand cost-effective flood forecasting model, however, it is questionable whether thisis achieved. In general, the improvement of model predictions is critical in reducingfuture material and immaterial damage caused by flooding. However, I found the relativecontribution of this study in the improvement of flood model predictions unclear.It is indistinct how significantly this study contributes to improvements in global floodpredictions. Whether the findings of this research are useful in other catchment areasover the globe is not defined.Overall, I found the methodology and the results of this study worked out well. Thestepwise approach is clear, and the representation of the results is very interestingto read. In the contrary, I found the introduction quite long, and very detailed. In myopinion, it difficult to find out what the broadly interesting knowledge gaps are. I suggestthe introduction to be focused on the importance and contribution of this study to theapplianceof flood model prediction on a global scale, and how that is achieved. Overall,this study fits wellwith the scope of the journal and can be published when severalrelatively minor issues are addressed. Below I provide more detailed comments.

  We thank Arnoud Goossen for his general assessment. We made the introduction more focused, highlighted the open science questions and clarified the objectives of the study. We further added a discussion on the steps still needed to enable global scale applications of the methodology introduced in this manuscript. In our opinion, the best way to evaluate the proposed methodology was to carry out a number of experiments in a controlled environment with synthetically generated and perfectly known data sets. While the results are encouraging and make it possible to envisage 'real-world' applications, it is also clear that more research is needed to get a better understanding of the adaptations that are required to systematically assimilate EO-derived flood extent observations into operational flood forecasting systems from around the world.

- Firstly, the introduction is very detailed on background information, but lacking in the focus of this specific research. The background information consists of a description of Data Assimilation (DA) and Synthetic Aperture Radar (SAR) images which is an essential part of the study. Different assimilation methods are discussed in detail, but it is not clear which and why this method is expected to be used. The detailed treatment of all the background information causes an unclear overview of what the study actually is about, as several DA methods are mentioned (KF, 4DVar, PF, EnKF (line 42-58).

  We agree that the introduction should be shortened and more focused. We argue that the studies of Revilla-Romero et al. (2016), Lai et al. (2014), and Cooper et al. (2019) are of paramount importance in the context of our work because they did not transform the information derived from SAR data into a state variable of the model prior to the

assimilation. To make the introduction more focused, we have condensed this part and removed unnecessary details about these methods:

"Different information about water extent can be extracted from a SAR image and used to improve the forecasts using DA techniques. Directly assimilating flood extent maps is not straightforward because these do not correspond to a state variable of the model. Therefore, some studies suggested to transform the SAR backscatter information into state variable prior to the assimilation. For instance, several studies have used EO-derived water levels to improve flood forecasts [e.g. Andreadis et al. (2007), García-Pintado et al. (2015), Matgen et al. (2010), Revilla-Romero et al. (2016), Giustariniet al.(2011), Hostache et al. (2010)]. The water levels are estimated by merging pre-selected flood extent limits extracted from the SAR satellite imagery with a digital elevation model (DEM). This step requires precise flood contour maps and high resolution DEMs which are not always available (Hostache et al., 2018). In the existing literature only a few studies have used DA for directly assimilating flood extent maps into flood forecasting models [e.g. Lai et al. (2014), Revilla-Romero et al. (2016), Cooper et al. (2018b), Cooper et al. (2018a), Hostache et al.(2018)]. Among the advantages of a direct use of the SAR backscatter values is that it reduces the data processing time that is a key-element in near-operational applications."

Then we move directly to the recent method of Cooper et al. (2018). This study, in which the information derived from the satellite is not transformed into a flood forecasting system variable, gives satisfying results in lines (cf. lines 49-53 of the revised untracked manuscript).

"Cooper et al. (2018a) have used an Ensemble Kalman Filter to update a 2D hydrodynamic model. In this case, the backscatter values are directly assimilated into the model without being transformed into state variables of the flood forecasting system. The dry and wet pixels of the simulated binary flood map are converted into equivalent SAR backscatter values corresponding to the spatial mean of the SAR backscatter observations. Cooper et al. (2018a) showed that the SAR backscatter-based assimilation method performs well compared to the assimilation method where the SAR backscatter is transformed into water levels."

From Line 67 up to line 70 of the revised tracked manuscript, some differences between PFs and other assimilation techniques were mentioned. As these differences have been widely reported and discussed in many papers, we decided to delete this part in order to condense the introduction and remove unnecessary details about these methods.

- The introduction does not lead to a specific research question or clear objective. The objective is mentioned twice (line 74 and line 91), but the choice of words is different, which causes confusion. The objective is mentioned, but a briefly description on how the objective is achieved is missing. The referencing of previous studies is used todiscuss this, but it is unclear which references are really used for the methodology.Now, only the last sentence of the introduction is stating that 'a sensitivity analysis of the DA framework with respect to the tempering coefficient is conducted', which is rather vague.

We fully agree that mentioning the objectives twice in a slightly different way created much unnecessary confusion. Therefore, we removed lines 87-88 of the revised tracked manuscript and condensed the objectives in one paragraph at the end of the introduction. The objectives are: evaluating the DA of Hostache et al. (2018) et al. in a controlled environment with synthetically generated data sets, carrying out a sensitivity analysis with

respect to the tempering coefficient and evaluating the effects on DA of errors in SAR observations.

"The main objective of the present study is to assess the strengths and the limitations of the DA framework previously proposedby Hostache et al. (2018). To do that we evaluate the DA framework in a fully controlled environment via synthetic twin experiments as this shall allow us drawing unambiguous and comprehensive conclusions. In addition, we conduct a sensitivity analysis of the DA framework with respect to the critical tempering coefficient that was recently introduced for tackling degeneracy more efficiently. We also aim to evaluate the effect of misclassified SAR pixels on DA. Therefore, errors are artificially added within the SAR image with the aim of getting a better understanding on how robust the proposed method is with respect to this type of errors. Results are evaluated not only locally but also over the entire flood domain and for subsequent time steps to the assimilation."

The tempering coefficient to reduce degeneracy is already introduced from line 68 to 72 of the revised untracked manuscript.

"Hostache et al. (2018) also highlighted that degeneracy may be a major issue of PFs: after the assimilation, the number of particles with high weights reduces to a few or only one particle so that the ensemble loses statistical significance. To overcomethis issue, Hostache et al. (2018) used a site-dependent tempering coefficient which inflates the posterior probability. In our study, we propose to adopt an enhanced tempering coefficient as a function of the desired effective ensemble size (EES) after the assimilation."

The reference used for the methodology, which is Hostache et al. (2018), is stated in line 66-68 of the revised untracked manuscript.

"The present study is a follow up of the study by Hostache et al. (2018) and carries out asimilar experiment in a controlled environment that considers the rainfall estimated together with SAR observations as the only source of uncertainty."

- For this major argument I would recommend being more specific with referring to previous studies, and to have a critical look on the broad background information. Especially, some of the referred studies concerning KF, 4DVar and PF seem unnecessary tome. Try to aim for a narrowing of the introduction, so that the introduction leads to theobjective and research questions of this study. I would remove the objective mentionedin line 74, as it is too subtle and not agree with the objective stated in line 91.

  To shorten and focus the introduction, we have removed from the manuscript the objective mentioned in lines 87-88 of the revised tracked manuscript as well as all unnecessary details on the DA techniques of the studies referred to in lines 67 up to line 70 of the revised tracked manuscript.

- The second argument concerns the conclusions of this study. Overall, I am very content with the conclusions. The conclusions of this research focus on the specific study area of the River Severn (UK), which is logically in line with the objective of this study. In line 376 it is stated that the main issue of using SAR observations in flood forecasting models is the difficulty of detecting flooded areas for specific cases, such as urban or vegetated areas. For now, this study only seems to be applicable for this specific study area, but I wonder if that is really the case. The societal significance of this study would be large if it contributes to a global improvement of flood modelling. I would recommend discussing the use of the findings of

this study on a global scale. As a reader I would like to know how these results improve the appliance of Sar observations for different types of land use, or if a significant error increase is expected when the analysis is performed for different types of landscapes or land uses.

To discuss the applicability of the method at large scale, we have added a paragraph in the discussion section of the revised paper (cf. lines 455-465 of the revised untracked manuscript).

"We also argue that the method used in the manuscript has the potential to support EO-based modelling at large scale. This potential is particularly high in large, natural floodplains where flood inundation remains present over long time periods. In spite of the increased frequency of satellite observations, the persistence of a flood over many days increases the chance of its detection and mapping by satellite sensors. Another condition that needs to be satisfied is that there should be an unambiguous relationship between the flood extent observed by the spaceborne sensors and river discharge. This also means that areas where backscatter variations are not impacted by the appearance of floodwater (e.g. densely vegetated floodplains) should be rather small. Indeed, these constraints must be satisfied to enable a successful application of the proposed framework and to take advantage of the analysis carried out in this manuscript. As a conclusion, based on the above elements, we argue that our approach is valid regardless of the type of model coupling that is performed and is thus applicable to many different forecasting systems. However, more research is needed to fully understand the role of floodplain and water basin characteristics and SAR data properties on the DA performance."

Regarding the error of SAR observations, as already mentioned in lines 402-407 of the revised untracked manuscript our results suggest that the DA can compensate for these kinds of errors if the percentage of misclassified pixels (i.e. in urban or vegetated areas) remains below 20% of the pixels of the SAR image. It means that model performances can improve if those areas are masked out or can be correctly classified.

"Our study further shows that it is important to characterize and mask out errors in the SAR observations. A large number of misclassified pixels substantially degrades the DA performance. In our case study, results suggest that an improvement of model simulations (i.e. water level and streamflow) in terms of CSI and RMSE performance metrics is achieved as long as errors in the observations are rather limited, i.e. when no more than 20% of the pixels are affected. However, if the misclassification goes beyond 40% of affected pixels, the assimilation has no effect and may even lead to a degradation of the model predictions."

- Minor argument 1: In line 68-69 it is mentioned that there could be other possible sources of uncertainty in the model system. I suggest some examples, as for now it is unclear in what direction these other sources uncertainties haveto be found.

It is true that other sources of uncertainty: "…input data, model parameters, initial conditions and model structure represent sources of uncertainty that affect the reliability and accuracy of flood forecasts" were only mentioned in the abstract (lines 2-3). We have added such examples also in lines 58-62 of the revised untracked paper.

"Forecast errors are reduced by a factor of 2 at the assimilation time and improvements persist for subsequent time steps up to 2 days. However, the improvements are not systematic: for some cases the updated hydraulic output deviates from the observations.

One of the reasons for such outliers could be the assumption that rainfall represents the dominating source of uncertainty together with satellite observation errors, thereby excluding other possible sources of uncertainty in the model system such as input data, model parameters, initial conditions and model structure."

- Minor argument 2: The spilling of water into the floodplain is modelled with a 2D diffusion wave scheme neglecting the convective acceleration (line 114). I would like to a reference or reasoning for the neglection of the convective acceleration. Even though it is logical, an assumption has been made about whether this term can be neglected, yes or no.

  The neglection of the convective acceleration for the simulation of the spilling of the water in the floodplain is rather common in order to simplify the shallow water equation. The 2D solver that has been used in the floodplain is the acceleration solver (Bates et al., 2010; De Almeida et al., 2012) which neglects only the convective acceleration. This reference has been added in lines 113-115 of the revised untracked manuscript.
  "When the storage capacity of the river is exceeded, the water spills into the floodplain and a 2D diffusion wave scheme neglecting the convective acceleration (de Almeida and Bates, 2013; Bates et al., 2010) is used for the floodplain flow simulation."

  In addition, the hydraulic model used here is based on the set-up defined in Melissa Wood et al. (2016) whose details are given in the Study area section (c.f. lines 281-283 of the revised untracked manuscript):
  "Channel width, channel depth, slope of terrain, friction of the flood domain and channel bathymetry are defined in each cell of the model domain as described in Wood et al. (2016). A uniform flow condition is imposed downstream. No lateral inflow in the hydraulic model is assumed."

- Minor argument 4: In this study, the alpha value is based on the desired effective ensemble size (EES) (line 222). It is unclear if this method has been used in previous studies and what the expected outcome of the use of this EES would be. I recommend adding some detail about this in the method section as it is a major change in comparison to the previous study by Hostache et al. (2018).
  The use of a tempering coefficient based on the effective ensemble size (EES) is commonly used in Particle Filters (van Leeuwen, 2019). The EES is used to obtain an idea of the number of particles having a non-negligible weight after the assimilation. However, the approach to compute $\alpha$ based on the EES is different from the study of Hostache et al. and to make this clearer we have changed sentences 228-229 of the revised untracked manuscript.
  "The coefficient $\alpha$ in Hostache et al. (2018) is site-dependent as it relies on the number of flood pixels, whereas in this study $\alpha$ is a function of the EES which is a measure of degeneracy based on the global weights (Arulampalam et al., 2002)"

- Minor issue 3: Line 216: "Since ï ¿A ¿a and weights have values are lower than one", missing "that" before "are"
  This has been corrected at line 222 of the revised untracked manuscript.
  "Since $\alpha$ and weights have values lower than one…"

- Methods Minor issue 2: Line 116: "No later inflow in. . .". I assume this is incorrect. Shouldn't this be 'latent inflow'
  This is a typo and it has been corrected in line 283-284 of the revised untracked manuscript.
  No lateral inflow in the hydraulic model is assumed.

- Results Minor issue 4: Line 274-276: It is unclear if this is correct. The revisit time is around 3-4 days, which means 2 satellite images per week. I do not understand how this results in four assimilated synthetic observations in a period of 10 days. 2 images per week ≈ 3 images per 10 days?

  In lines 288-291 of the revised untracked manuscript we have added the orbit details in order to be clearer on the acquisition frequency.

  "The virtual satellite acquisition dates are aligned with the actual Sentinel-1 acquisition frequency. The revisit time over Europe, considering both ascending and descending orbits, is around 3-4 days meaning that on average 2 satellite images are available per week. In order to adopt a realistic Sentinel-1-like observation scenario we chose to assimilate four synthetic observations over a period of 10 days."

- Minor issue 5: Fig 3 & 4: I have not read what the pixel size of the SAR observation is. I think it is important to mention the pixel size of the SAR observations by the Sentinal-1 satellite.

  The SAR resolution is provided in the revised manuscript and it has the same resolution of the LISFLOOD-FP maps as mentioned in line 88 of the revised untracked manuscript.

  "The SAR data are synthetically generated with a pixel spacing of 75 m."

- Minor issue 6: Fig 6: Labels incorrect. In the graph of streamflow time series (left) the assimilation of image I is indicated four times, while in the image on the right the labels assimilation of image I, II, III, IV are given.

  The figure 6 has been corrected in the revised version.

- Minor issue 7: Line 301 & Fig 7: "higher than the orange ones. . .". As the comparison between over detection (red) and under detection (black) is probably meant, "orange" should be "black

  This has been corrected in the revised version.

**2. Anonymous Referee #1**

- The paper addresses a topic of immense community interest. The methodological design is sound, and the overall writing quality is quite good. However, this paper lacks scientific/conceptual contribution.

  The main contribution of this paper is rather technical. I say this because the concept of assimilating remotely sensed flood maps into flood models is not new. While the authors nicely rationalized their limited focus (by clarifying that their goal is to assess previous DA frameworks and draw generic conclusions; see P4), I see a major conceptual issue which may put this paper in a "conflicting" position against the current state of science. See below.

  We thank Referee #1for the assessment that made us re-think some critical aspects of our study.

  The methodology presented in this paper is not applicable to the common practice offlood inundation modelling/forecasting. Specifically, regardless of DA technique (e.g.,particle filter), effect of SAR observations cannot be fed back to streamflow and stageheight unless the hydrology and hydrodynamic models are tightly coupled. Most of thelarge basin/continental-scale flood inundation forecasting frameworks rely on looselycoupled hydrology (A) and hydraulic (B) model components. In such a framework,there is only a one-way transfer of information from A to B using a relational datamodel(Peckham et al., 2013). The VIC and LISFLOOD-FP coupling by Schumann etal. (2013), the VIC, Delft3D, and LISFLOOD-FP coupling (GLOFRIM framework) byHoch et al. (2017), and the more recent SWAT and LISFLOOD-FP coupling by Rajib etal. (2020), all rely on loose coupling of models;

as such, the approach presented in thispaper (and the underlying math) cannot be generalized. Therefore, I strongly suggestadding a separate paragraph in the introduction highlighting this limitation (dear authors:please feel free to recycle the above texts and references when you revise yourpaper).

Thank you very much for suggesting this clarification and providing relevant references. We would argue that one of the strengths of the proposed DA method is that it can be applied to different loosely coupled flood forecasting frameworks as it enables via a post-processing of the model data a feedback from SAR observations to streamflow and stage height simulations. The weights are used to calculate the expectation of water levels and discharge at the assimilation time and for subsequent time steps. No state variable of the hydrological model or hydraulic model is updated. Therefore, we argue that even though the models are loosely-coupled (one-way transfer of information), it is possible to apply the proposed methodology also to different model forecasting chains. Within the discussion section we have added a paragraph (lines 441-454 ofthe revised untracked version) of the manuscript to take into account the comments and the references suggested by Referee#1.
"Our DA framework can be applied to a variety of flood inundation forecasting chains. In fact, the forecast updating is carried out via a sequential importance sampling only (i.e. importance weights). Only the particle weights are updated based on the observations and used to compute the expectation (i.e. weighted mean) of the augmented state vector including hydraulic state variables of water depth, plus flood extent and boundary conditions. In this study the hydrologic and hydraulic models are loosely coupled with a one-way transfer of information as in many other studies [e.g., Peckham et al. (2013), Hoch et al. (2017), Rajib et al. (2020)]. The weights define the relative importance of the particles and thus of the inherent streamflow and stage along the entire river. We acknowledge that the observed flood extent is more closely linked to the past boundary conditions rather than the boundary conditions corresponding to the assimilation time steps. In spite of this limitation we argue that in this synthetic experiment, the particles that performed best in the past are also those that reach the highest performance level at the time of the assimilation. This is illustrated in the Figures 10 and 11 where the use of updated weights is shown to enable the correction of the state variables of the hydraulic model both upstream and downstream. However, we recognize that further improvements could be developed to address issues such as spurious relations that may occur between SAR observations and model variables due to a rather small ensemble size. Enlarging the ensemble size could be necessary if this occurs."

- Accordingly, I also recommend editing the title as "Assimilation of probabilistic flood maps from SAR data into a coupled hydrologic-hydraulic forecasting model: a proof of concept".

Thanks for that suggestion. We have changed the title accordingly.

**3. Anonymous Referee #2**

- **This is a highly technical manuscript focused on assimilation of many differentdata sources using multiple techniques to predict flood extent and depth.I think this is an interesting study, but I overall I think it needs major improvementsbefore it can be published. The science is sound and interesting, but themanuscript could be clarified and re-vised throughout to make this easier for the reader to understand. I summarize my major comments and minor commentsbelow.**

- My main recommendation to the authors is to clearly clarify the contributionof this study to the literature. The manuscript incorporates many technicalmethodological

assessments, but it is not always clear why these assessments are being conducted, and what they help us learn about flood modelling. The authors should clearly state their contributions in the introduction and clarify in a discussion section how their findings advancethose conducted by other studies.The introduction should be revised and reorganized. At current, theintroduction is very technical, and describes a lot of the existing literature.However, I had a hard time following the common threads and major pointsbeing made across the arc of the introduction. Many individual referencesare described but aren't necessarily connected to the bigger picture offlood modelling.

We thank Referee #2 for the detailed review.

The introduction has been shortened and clarified in accordance with the referee's recommendations. Less details on the existing literature are given and more emphasis is put on the open science questions and the objectives of our study (lines 37 – 48 of the revised untracked manuscript).

"Different information about water extent can be extracted from a SAR image and used to improve the forecasts using DA techniques. Directly assimilating flood extent maps is not straightforward because these do not correspond to a state variable of the model. Therefore, some studies suggested to transform the SAR backscatter information into state variable prior to the assimilation. For instance, several studies have used EO-derived water levels to improve flood forecasts [e.g. Andreadis et al. (2007), García-Pintado et al. (2015), Matgen et al. (2010), Revilla-Romero et al. (2016), Giustarini et al.(2011), Hostache et al. (2010)]. The water levels are estimated by merging pre-selected flood extent limits extracted from the SAR satellite imagery with a digital elevation model (DEM). This step requires precise flood contour maps and high resolution DEMs which are not always available (Hostache et al., 2018). In the existing literature only a few studies have used DA for directly assimilating flood extent maps into flood forecasting models [e.g. Lai et al. (2014), Revilla-Romero et al. (2016), Cooper et al. (2018b), Cooper et al. (2018a), Hostache et al.(2018)]. Among the advantages of a direct use of the SAR backscatter values is that it reduces the data processing time that is a key-element in near-operational applications."

Moreover, we agree that the paragraph in lines 58-61 of the unrevised manuscript could create some confusion as it is not completely aligned with the rest of the introduction and therefore it has been removed. With these adjustments, we believe that the reader can better follow the main reasoning and the major points of the introduction.

- More synthesis is needed across these references and paragraphs to highlightthe major knowledge gaps. Furthermore, I'd recommend shorteningthe introduction.
  We have removed details of Revilla-Romero et al. (2016), Lai et al. (2014) in order to shorten the introduction. We have also removed details on the difference of the principal DA methods in lines 58-61 of the unrevised manuscript.

- Finally, the introduction section normally concludes with a statement aboutthe novelty of the study, the scope, and the objectives. These are instead first introduced on line 75, then again later in the introduction. I'd recommendconsolidating these statements into a coherent paragraph at the endof the introduction.
  We fully agree with this assessment (see also our answer to a similar comment from Referee #1). The objectives are now condensed in one paragraph at the end of the introduction section line (78-85 of the revised untracked manuscript).
  "The main objective of the present study is to assess the strengths and the limitations of the DA framework previously proposed by Hostache et al. (2018). To do that we

evaluate the DA framework in a fully controlled environment via synthetic twin experiments as this shall allow us drawing unambiguous and comprehensive conclusions. In addition, we conduct a sensitivity analysis of the DA framework with respect to the critical tempering coefficient that was recently introduced for tackling degeneracy more efficiently. We also aim to evaluate the effect of misclassified SAR pixels on DA. Therefore, errors are artificially added within the SAR image with the aim of getting a better understanding on how robust the proposed method is with respect to this type of errors. Results are evaluated not only locally but also over the entire flood domain and for subsequent time steps to the assimilation."
Objectives in line 75 of the unrevised manuscript have therefore been removed.

- At the end of the introduction, I am left unsure of the scope and objectivesof the manuscript (for instance, nothing about SAR or flooding ismentioned). These three concluding sentences could benefit from morespecifics as to what will be tested and explored in this particular article.Specifics, such as types of model used, data resolution, etc could be specifiedhere, to more clearly articulate to the reader the framing of your particularwork.
We have added a paragraph where specifics such as types of model used, data resolution are mentioned (lines 85-88 of the revised untracked manuscript). More details are given in the methods section.
"To carry out the experimental study we apply the DA framework to a forecasting system consisting of a loosely-coupled hydrological model (SUPERFLEX) and hydraulic model (LISFLOOD-FP). The meteorological data that areused to run the experiments are derived from the ERA-5 archive with a spatial resolution of 25 km and a temporal resolutionof 1 hour. The SAR data are synthetically generated with a pixel spacing of 75 m."

- The methods section is very detailed (which I appreciate). Yet, I had ahard time understanding the major comparisons to be made in the results/discussion section Could you more clearly summarize these and whyyou are comparing these methods at the start of this section?
We have added the following paragraph from line 98 to line 103 in the revised untracked version of the manuscript.
"The three conducted experiments are summarized as follows:
(a) An application of the standard PF where degeneracy occurs;
(b) An application of the adapted PF where a tempering coefficient is used to reduce degeneracy. We also investigated the sensitivity of the DA results to different values for the tempering coefficient, corresponding to effective ensemble sizes of 5, 10, 20 and 50%;
(c) An application of both proposed methods with artificially introduced known errors into the SAR image classification in order to evaluate the impact of these errors on the DA performance metrics."

- The workflow is helpful, but with the number of methods and acronyms, I had a hard time following this.
We have removed the acronyms in the flow chart caption of Figure 1 to make it more easily readable:
"Flow chart of the synthetic experiment. The true rainfall is perturbed. The same flood forecasting model structure composed of a hydrological model and a hydraulic model is used to obtain the probabilistic flood map and the ensemble of binary flood maps. The

probabilistic flood map is assimilated into the ensemble of binary flood maps via the Particle Filter to obtain the weights with which the expectation of water levels, streamflow and flood extent are computed."

- The Study Area section comes after the methods section – this was a littleconfusing to me, because the nuances of this are discussed in the methodssection. Is it worth switching the order of these?
  "Channel width, channel depth, slope of terrain, friction of the flood domain and channel bathymetry are defined in each cell of the model domain as described in Wood et al. (2016). A uniform flow condition is imposed downstream. No lateral inflow in the hydraulic model is assumed."
  This paragraph has been moved into the section Study area of the revised paper. However, we kept the study area section after the methods section because we want to give a more general overview of the methodology (not related to the study area) and to show that it is applicable to many cases, whereas the study area is more specific, closer to our particular situation.

- Please ensure that all paragraphs are 3+ sentences and ensure that theseare appropriately combined throughout the text.
  This has been checked and corrected in the revised version of the manuscript.

- I would recommend relabelling sub-sections within the results to separateout the different comparisons and techniques you are making – organizingthese headings would help me connect what you do to your methods section. For instance, I had a hard time connecting these results to the stated objectiveof detecting uncertainty in precipitation, and then to the conclusionssection. It could also help to start each sub-section by describing whatmethods/approaches you are testing and why, given there are many comparisons.
  Titles for each subsection of the results section have been changed according to reviewer's suggestion, as follows:

  4.1   Synthetic SAR and ensemble generation and evaluation
  4.2   Evaluation of the flood extent map estimated at the assimilation time
  4.3   Evaluation of the flood map estimated in time
  4.4   Evaluation of the water levels in time over a global scale
  4.5   Evaluation of discharge and water level time series
  4.6   Impact assessment of errors in SAR observations

- This may be a personal preference, but I would recommend shortening theconclusions section, and moving much of what is in there now to a discussionsection.
  We have modified the discussion and conclusion sections in the revised paper following the recommendation of the reviewer.
- Within this discussion section, the main piece I don't see is a discussion ofthe limitations of this approach.
  Limitations are now defined in the discussion section of the revised manuscript as indicated below.
- Is there a reason to think that this approach is transferable? Why orwhy not?
  This has been changed in lines 441-454 of the revised untracked manuscript:
  "Our DA framework can be applied to a variety of flood inundation forecasting chains. In fact, the forecast updating is carried out via a sequential importance sampling only (i.e. importance weights). Only the particle weights are updated based on the observations

and used to compute the expectation (i.e. weighted mean) of the augmented state vector including hydraulic state variables of water depth, plus flood extent and boundary conditions. In this study the hydrologic and hydraulic models are loosely coupled with a one-way transfer of information as in many other studies [e.g., Peckham et al. (2013), Hoch et al. (2017), Rajib et al. (2020)]. The weights define the relative importance of the particles and thus of the inherent streamflow and stage along the entire river. We acknowledge that the observed flood extent is more closely linked to the past boundary conditions rather than the boundary conditions corresponding to the assimilation time steps. In spite of this limitation we argue that in this synthetic experiment, the particles that performed best in the past are also those that reach the highest performance level at the time of the assimilation. This is illustrated in the Figures 10 and 11 where the use of updated weights is shown to enable the correction of the state variables of the hydraulic model both upstream and downstream. However, we recognize that further improvements could be developed to address issues such as spurious relations that may occur between SAR observations and model variables due to a rather small ensemble size. Enlarging the ensemble size could be necessary if this occurs."

In what scenarios is this approach more useful (i.e. at what scale)?
This has been changed in lines 455-465 of the revised untracked manuscript.
"We also argue that the method used in the manuscript has the potential to support EO-based modelling at large scale. This potential is particularly high in large, natural floodplains where flood inundation remains present over long time periods. In spite of the increased frequency of satellite observations, the persistence of a flood over many days increases the chance of its detection and mapping by satellite sensors. Another condition that needs to be satisfied is that there should be an unambiguous relationship between the flood extent observed by the space-borne sensors and river discharge. This also means that areas where backscatter variations are not impacted by the appearance of floodwater (e.g. densely vegetated floodplains) should berather small. Indeed, these constraints must be satisfied to enable a successful application of the proposed framework and to take advantage of the analysis carried out in this manuscript. As a conclusion, based on the above elements, we argue that our approach is valid regardless of the type of model coupling that is performed and is thus applicable to many different forecasting systems. However, more research is needed to fully understand the role of floodplain and water basin characteristics and SAR data properties on the DA performance."

- Given that rainfall is the main source of uncertainty, what does thismean for future work?
This has been explained in lines 409-416 of the revised untracked manuscript.
"The results of our study confirm the effectiveness of the proposed DA framework when the hypothesis of the rainfall as the main source of uncertainty is verified. Consequently, for those cases where rainfall represents the main source of uncertainty, more obviously but not only in poorly and un-gauged catchments and when using medium-range forecasting models, our study results indicate that the application of the approach described in the manuscript may lead to improved results of the model simulations. For those cases where the uncertainty of other sources becomes more relevant and may be even dominant, it is clear that such sources need to be taken into account explicitly. However, the required adaptations of the proposed DA framework still need to be developed. In this context it is also worth mentioning that the limitations identified in the previously published real case study by Hostache et al. (2018) were explained by additional sources of uncertainties not taken into account."

- Can this work improve forecasting?

  This has been explained in lines 417-423 of the revised untracked manuscript.

  "Using probabilistic flood maps or backscatter values increases the number of observations to be assimilated when compared to a method that only derives the flood edge from satellite observations as reported in Cooper et al. (2018a). Moreover, the nearly direct use of the SAR information enables a faster end-to-end processing from the acquisition of the image to the assimilation of the SAR data into the model which is beneficial for an operational usage. In our experiments, the improvements of model forecasts of water level and streamflow are significant at the assimilation time step and the improvements persist over subsequent time steps (for example up to 27 hours after the first assimilation the model results outperform the open loop simulation)."

- Line 42 – "used" is repeated Line 48-49 – I had trouble understanding thissentence – could you rephrase? It was not clear to me what 'the latter' referredto:

  The sentence has been rephrased in lines 37-38 of the revised untracked manuscript.

  "Directly assimilating flood extent maps is not straightforward because these do not correspond to a state variable of the model. Therefore, some studies suggested to transform the SAR backscatter information into state variable prior to the assimilation."

- Line 42 – I am missing the connection from this paragraph to the next – whywould one want to use a KP, 4DVar, or PF technique for assimilation of flood information? Can you connect these thoughts to the previous sentence?

  We have removed details on the techniques used in the different cited studies.

- Line 52 – is there a reason to have a whole paragraph focused on this particulararticle? Is it most similar to what is done in this study? Do youimprove on their work? If not, I'd recommend shortening the description ofthis article:

  We have removed unnecessary details of Cooper et al (2018). The sentence has been rephrased in lines 49-53 of the revised untracked manuscript.

- "Cooper et al. (2018a) have used an Ensemble Kalman Filter to update a 2D hydrodynamic model. In this case, the backscatter values are directly assimilated into the model without being transformed into state variables of the flood forecasting system. The dry and wet pixels of the simulated binary flood map are converted into equivalent SAR backscatter values corresponding to the spatial mean of the SAR backscatter observations. Cooper et al. (2018a) showed that the SAR backscatter-based assimilation method performs well compared to the assimilation method where the SAR backscatter is transformed into water levels."

- Lines 76 – 90 – this is very detailed, to the point where I am unsure if this ishelpful in the introduction. Would you be able to shorten this section anddistil a few key messages? Could this be moved to the methods section?

  The paragraph has been restructured and shortened. Some relevant details have been moved to the methods section.The following paragraph remains in the introduction:

  "Moreover, in Hostache et al. (2018), speckle errors in the SAR observations, are taken into account through the Bayesian approach introduced by Giustarini et al. (2016). However, no conclusions are drawn concerning the effect of misclassified pixels. In fact, for some particular cases such as densely vegetated areas, the detection of floodwater from SAR imagery is known to be prone to errors. Detecting and removing such errors represents one of the main scientific challenges of using SAR data for a systematic, fully automated, large-scale flood monitoring (and prediction)."

  The following paragraph has been moved to the methods section:

  "The method proposed by Giustarini et al. (2016) aims at characterizing the speckle-induced uncertainty. However, it does not consider any other phenomena leading to a

wrong classification in SAR-based flood maps. Particular atmospheric conditions(e.g. wind, snow, precipitation), water-look-a-like areas (e.g. asphalt, sand, shadow) or obstructing objects (e.g. dense canopy,buildings), as mentioned in Giustarini et al. (2015), can lead to a wrong classification in the flood maps. Therefore, the areas where such errors could occur should be masked out from the SAR-based flood maps in order to provide a reliable flood detection."

- Line 178: "supposed to be uniform" – do you mean assumed to be uniform?Sampled as uniform? Please clarify:
  The sentence has been corrected.

- Section 2.3 – please weave the equations into the text, instead of listingthem after the text here:
  This has been done in the revised manuscript.

- Section 2.4 – please do a thorough read to ensure that all variables in thecontained equations are clearly defined in this section:
  This has been changed. The "δ is the Dirac delta function" was missing.

- Lines 228 – 232 – this reads as 'results' – should this be moved to the resultssection?:
  The paragraph has been moved to the conclusion section of the revised manuscript.
  "In this study, we have evaluated the effect of variations of the tempering coefficient on the DA performance. Different PFs are compared with the OL and the synthetic truth: the SIS (with only a few particles from the ensemble potentially carrying non-negligible weights) and the adapted method with 5-10-20-50% EES (with the number of particles with non-negligible weights increasing with the EES). This methodology leads to slightly biased estimates because the observation is down-weighted."

- Section 3.0 – please capitalize 'area':
  The area word in the title has been capitalized

- Line 276: The plots in this section show four time points –why did you selectthese time points? Please introduce the time points in this section:
  In lines 288-291 of the revised untracked manuscriptare explained.
  "The virtual satellite acquisition dates are aligned with the actual Sentinel-1 acquisition frequency. The revisit time over Europe, considering both ascending and descending orbits, is around 3-4 days meaning that on average 2 satellite images are available per week. In order to adopt a realistic Sentinel-1-like observation scenario we chose to assimilate four synthetic observations over a period of 10 days."

- Line 285: You show a sub-section of the result area multiple times – pleaseintroduce this area and why you selected it in the text. Also – are you computingresults for just this section of the river or the entire watershed? Iwasn't sure from the methods and study area section. Please clarify:
  Further clarifications are reported in the results section lines 292-294 of the revised untracked manuscript are explained.
  "In the Figures 3 and 4, the area corresponds to the hydraulic model domain. The hydrologic model, covering the upstream catchment, is used to compute the input boundary conditions of the hydraulic model. Results are computed and compared within the hydraulic model domain. "

- Line 274 – 284 – should this be in methods?
  This paragraph has been moved to the beginning of the methods section of the revised version.

- 279 – 281 – what is the significance of this? Could you explain more why you mention this here? This again seems like 'methods' – should this be moved to the methods section, or is it a 'result' of your investigation?
  Lines 279 – 281 have been moved to the methods section from line 141-151 of the revised untracked manuscript.
  "While this study is based on a synthetic experiment, true binary flood extent maps are available. Therefore, the assimilation is realized using both the estimated prior probability (as the ratio between the flooded area and the total area) and the prior probability equal to 0.5. Given the similarity of the results for both cases, in the following sections we only discuss the experiment using the true prior probability. The method proposed by Giustarini et al. (2016) aims at characterizing the speckle-induced uncertainty. However, it does not consider any other phenomena leading to a wrong classification in SAR-based flood maps. Particular atmospheric conditions (e.g. wind, snow, precipitation), water-look-a-like areas (e.g. asphalt, sand, shadow) or obstructing objects (e.g. dense canopy, buildings), as mentioned in Giustarini et al. (2015), can lead to a wrong classification in the flood maps. Therefore, the areas where such errors could occur should be masked out from the SAR-based flood maps in order to provide a reliable flood detection."

- Line 284 – Figure 3 and Figure 5 are mentioned – figures should be listed in order. Figure 4 is not cited in the text. Should this be removed or moved to Supporting Information?
  The typo has been corrected.

- Line 315 – Please do not start a sentence with a number:
  This has been corrected.

- Line 387 – 399 – could you rephrase this sentence? I don't understand what it is saying:
  - Paragraph (a) "Although the use of a smaller tempering coefficient leads to a larger effective ensemble size (e.g. 50 % ) and helps avoid degeneracy, the results are less accurate compared to the standard method or a 5% EES method" has been rephrased and put in lines 434-439 of the revised untracked manuscript
    "Some modifications of the DA framework are still required to fully overcome the issue of degeneracy. Although the use of a smaller tempering coefficient leads to a larger effective ensemble size (e.g. 50 %) and helps avoid degeneracy, the results are less accurate compared to the standard method or the adapted method with 5% EES. As described in Neal (1996) and in van Leeuwen et al. (2019), the tempering procedure consists of several steps, but in this study the tempering coefficient is applied only to flatten the likelihood, therefore down weighting the observations. This most likely explains why the data assimilation performs better when the effective ensemble size (the number of particles not negligible after the assimilation) is smaller."
  - Paragraph (b) "The persistence in time of the beneficial effects of the assimilation varies according to the rapidity of variations of flood extent; a more frequent image acquisition could help in better keep the predictions on track." has been rephrased and put in lines 421-433 of the revised untracked manuscript
    "In our experiments, the improvements of model forecasts of water level and streamflow are significant at the assimilation time step and the improvements persist over subsequent time steps (for example up to 27 hours after the first assimilation the model results outperform the open loop simulation). The persistence of these improvements depends on the flashiness of the flood event (i.e., the rapidity with which hydrologic conditions change). More frequent image

acquisitions could help keep model predictions on track, especially when the system is highly dynamic. The update of a state variable of the forecasting model could as well increase the persistence of the improvements. In our study none of the model state variables is updated as only the particle weights are computed, based on the SAR observations and on the simulated flood extent maps and used to calculate the expectation of water levels and streamflow. In previous studies [Andreadis et al. (2007), Matgen et al. (2010), Cooperet al. (2018b)], inflow updating was identified as a condition leading to more persistent improvements. For instance, one ofthe conclusions from the study by Matgen et al. (2010) was that updating the fluxes at the upstream boundary conditions, rather than the water levels, is more effective because of the high uncertainty of the inflow due to the poorly known rainfall distribution over the catchment. Therefore, as a future perspective, we aim to update hydrologic model states because it might have a positive impact on the long-term runoff simulations and consequently on the persistence of DA benefits."

- Paragraph (c) "Our study further shows that it is important to characterize and mask errorsin the SAR observations. A large number of misclassified pixels substantiallydegrades DA performance. In our study, the improvement of model simulations (water levels and streamflow) and performances (CSI and RMSE) after the assimilation is still possible if the errors in the SAR observations are rather limited (not more than the 20% of the pixels). However, if the misclassification goes beyond 40% of the pixels, the assimilation has no effect oreven degrades the model predictions"has been rephrased and putin lines 402-407 of the revised untracked manuscript.

- "Our study further shows that it is important to characterize and mask out errors in the SAR observations. A large number of misclassified pixels substantially degrades the DA performance. In our case study, results suggest that an improvement of model simulations (i.e. water level and streamflow) in terms of CSI and RMSE performance metrics is achieved aslong as errors in the observations are rather limited, i.e. when no more than 20%of the pixels are affected. However, if the misclassification goes beyond 40% of affected pixels, the assimilation has no effect and may even lead to a degradationof the model predictions."

- Paragraph (d) "The results confirm the validity of the DA framework when the hypothesis of the rainfall as main source of uncertainty is verified. This confirms that the limitations identified in the previous real case study by Hostache et al. (2018) could be explained by additional sources of uncertainties that were not taken into account" has been rephrased and put in lines 409-416 of the revised untracked manuscript

  "The results of our study confirm the effectiveness of the proposed DA framework when the hypothesis of the rainfall as the main source of uncertainty is verified. Consequently, for those cases where rainfall represents the main source of uncertainty, more obviously but not only in poorly and un-gauged catchments and when using medium-range forecasting models, our study results indicate that the application of the approach described in the manuscript may lead to improved results of the model simulations. For those cases where the uncertainty of other sources becomes more relevant and may be even dominant, it is clear that such sources need to be taken into account explicitly. However, the required adaptations of the proposed DAframework still need to be developed. In this context it is also worth mentioning that the limitations identified in the previously published real case

study by Hostache et al. (2018) were explained by additional sources of uncertainties not taken into account."

- Figure 2: Could you highlight on this figure the places you select for Figure3 and Figure 4?
  A black square has been drawn in figure 2 corresponding to the domain of the hydraulic model of figures 3 and 5.

- Figure 3 and 4: The legend is hard to see, and there is nolabel of what 'value' is being shown (and its associated units).
  This has been modified.

- Figure 3 and4: What are the four assimilation time steps? Please label these figures as(a)-(d) or on the figure to indicate this.
  It has been done in the revised version of the manuscript.

- Figure 3 and 4: Should these becombined to enable comparison? It is not entirely clear from the resultstext what these images show and how these connect to the workflow.
  It is explained in lines 294-296 of the revised  untracked version of the manuscript.
  "The synthetic SAR observations are shown in Figure 3. The corresponding PFMs are shown in Figure 4 and reliability plots are provided in Figure 5. In the reliability plots, the points aligned along the 1:1 line indicate a statistically reliable PFM. "

- Table1 and Table 2: Please direct readers to Figure 6 in the captions for these:
  The caption of table 1 and 2 have been corrected.